There are amendments to this paper

# The future of Southeast Asia's forests

Ronald C. Estoque[1], Makoto Ooba[1], Valerio Avitabile[2], Yasuaki Hijioka[1], Rajarshi DasGupta[3], Takuya Togawa[1] & Yuji Murayama[4]

While Southeast Asia's forests play important roles in biodiversity conservation and global carbon (C) balance, the region is also a deforestation hotspot. Here, we consider the five shared socioeconomic pathways (SSPs) to portray a range of plausible futures for the region's forests, employing a state-of-the-art land change modelling procedure and remotely sensed data. We find that by 2050 under the worst-case scenario, SSP 3 (regional rivalry/a rocky road), the region's forests would shrink by 5.2 million ha. The region's aboveground forest carbon stock (AFCS) would decrease by 790 Tg C, 21% of which would be due to old-growth forest loss. Conversely, under the best-case scenario, SSP 1 (sustainability/taking the green road), the region is projected to gain 19.6 million ha of forests and 1651 Tg C of AFCS. The choice of the pathway is thus critical for the future of the region's forests and their ecosystem functions and services.

[1] National Institute for Environmental Studies, Tsukuba, Japan. [2] European Commission, Joint Research Centre (JRC), Ispra, Italy. [3] Institute for Global Environmental Strategies, Kanagawa, Japan. [4] University of Tsukuba, Tsukuba, Japan. Correspondence and requests for materials should be addressed to R.C.E. (email: estoque.ronaldcanero@nies.go.jp) or (email: rons2k@yahoo.co.uk)

Tropical forests occupy only about 7% of the earth's land surface but are home to nearly two-thirds of the world's floral and faunal diversity[1,2]. They play a pivotal role in global carbon (C) balance and climate change mitigation[3], storing 228.7 Pg C in their woody vegetation[4] and accounting for 68% of global C stock[5]. They also contribute to ecosystem-based adaptation, a concept that relates to the use of biodiversity and ecosystem services in an overall climate adaptation strategy[6,7]. Yet, there has been an unprecedented loss of tropical habitats, owing to a multitude of anthropogenic activities[1,8–15]. In fact, tropical deforestation is responsible for around one-tenth of total anthropogenic C emissions[5]. Most significantly, while they were once considered a moderate sink for atmospheric C[13], a recent study indicated that the C balance of tropical forests has tilted towards net source[14] due to extensive deforestation and a reduction in C density[14,16].

Southeast Asia is home to nearly 15% of the world's tropical forests[17], and includes at least four of the twenty-five globally important biodiversity hotspots[8]. The region, however, is also among the world's major deforestation hotspots, responsible for the bulk of deforestation in tropical humid and low-land forests[8,9,18–20]. Estimates suggest that habitat loss in Southeast Asia is among the highest[9,19] and most severe in terms of biodiversity loss[8,11], while deforestation rate is comparable only to that of Latin America[19]. Between 1990 and 2010, Southeast Asia registered an average net loss of 1.6 million ha yr$^{-1}$ (0.6% yr$^{-1}$), reducing the region's forest cover from 268 million ha to 236 million ha[17]. Given these rates, and the fact that over 90% of Southeast Asia's forests were still unprotected in the early 2000s[8], it is feared that over 40% of the region's biodiversity may vanish by 2100[11].

Numerous studies have indicated that some of Southeast Asia's intact forests (IFs) and protected areas (PAs), which are among the prime reserves for tropical biodiversity and aboveground forest carbon stocks (AFCS), have been degraded and converted to non-forest purposes[1,8,20]. Forest clearance and canopy loss in the region are attributed to several large-scale, anthropogenic drivers, including logging and clear-cutting for food production, cash crops and agriculture[8,11,17,20–22]. Yet, it is difficult to generalise these drivers owing to the diverse demographic, economic and policy settings of the respective countries. Moreover, while it is evident that uncontrolled exploitation and degradation will continue to affect the future state of the region's forests, the future might not be entirely bleak. In fact, over the last decade there have been significant indications of favourable landscape changes leading to afforestation and forest regrowth[17,22].

Considering the multiple interacting uncertainties and the dynamics of socioeconomic systems, charting the path of the region's forest future is a challenge, and requires exploratory scenario-based analyses. Scenario analysis is a useful technique for assessing the social and ecological impacts of interventions, and is thus an integral part of planning and decision-making[23–26]. Scenario analysis is a structured process of exploring and evaluating alternatives aimed at providing insights regarding plausible rather than probable futures[23,24,27].

In this study, we aimed to develop spatially explicit forest cover change scenarios for Southeast Asia and to monitor potential future forest cover changes and their consequent environmental impacts. Our approach is based on the recently formulated shared socioeconomic pathways (SSPs), a new generation scenario framework developed in coherence with future radiative forcing (the representative concentration pathways or RCPs) and their associated socioeconomic trajectories[28,29]. These SSPs provide significant advances from the previous scenario frameworks, especially the Intergovernmental Panel on Climate Change (IPCC) Special Report on Emissions Scenarios (SRES)[30], by considering the uncertainty space in mitigation and adaptation challenges. The five SSPs outline different storylines and assumptions of global development pathways and focus on qualitative descriptions of likely future changes in demographics, human development, economy and lifestyle, policies and institutions, technology, and land use and forest resources[26,28,29,31,32].

The five SSPs are: SSP 1, or the sustainability/taking the green road scenario; SSP 2, or the middle of the road scenario; SSP 3, or the regional rivalry/rocky road scenario; SSP 4, or the inequality/ road divided scenario; and SSP 5, or the fossil-fuelled development/taking the highway scenario[26,28,29,31,32]. Of these five scenarios, SSP 1 presents the highest increase in forest cover in the Asian region (one of the five SSP regions) from 2015 to 2050, and SSP 3 presents the greatest decrease (see Methods). SSP 1 assumes inclusive development and respect for perceived environmental boundaries, as well as high investment in human capital, education and awareness[28]. Conversely, SSP 3 assumes fragmentation, comparatively weak global institutions and a lack of cooperation in addressing global environmental concerns, together with poor investments in education and awareness[28]. In a nutshell, these two scenarios represent the two opposing ends of the scenario spectrum, in which SSP 1 signifies low mitigation and adaptation challenges, whereas SSP 3 signifies the opposite.

The SSPs, however, are primarily designed for global scale projections and analyses, and although forest cover change projections are available, they are limited to quantities and have no spatial dimension. There is thus a need to spatially allocate these projected forest cover changes in order to facilitate impact analysis and, at the same time, support local studies and environmental monitoring.

In this study, we have spatially allocated the projected future forest cover changes under the five baseline SSPs by employing a state-of-the-art land change modelling approach and using remotely sensed data (2015–2050). We examined the potential implications of these spatially allocated forest cover changes by quantifying their consequent AFCS changes at the country and province levels, across forest cover classes, and within the IFs and PAs in Southeast Asia, given their important roles as prime reserves for tropical biodiversity and AFCS (see Methods).

## Results

**Past-to-present forest and carbon stock losses.** Southeast Asia lost about 80 million ha of forest between 2005 and 2015, which translates to a forest loss rate of around 8 million ha yr$^{-1}$ (Supplementary Table 1). Of this forest loss, Indonesia accounted for almost two-thirds with a 62.0% share. Malaysia came second with a 16.6% share. The other top forest-losing countries in the region were Myanmar and Cambodia, with respective shares of 5.3 and 5.0%. The region therefore lost a total of 998 Tg C AFCS (year 2000 equivalent) during the 2005–2015 period, equivalent to a loss rate of around 100 Tg C yr$^{-1}$. Indonesia was also the largest contributor with a 62.1% share. Malaysia had a share of 17.4%, while Cambodia and Myanmar had 5.3 and 4.6%, respectively.

**Projected changes in forest cover and carbon stock.** Southeast Asia was covered with 206.5 million ha of forest in 2015, containing a total of 21,172 Tg C AFCS (Figs. 1 and 2). Indonesia is the largest contributor both in terms of forest cover and AFCS at 56 and 65%, respectively (Fig. 2). Among the five SSPs, SSP 1 would result in the highest net forest cover gain by 2050, with 19.6 million ha, followed by SSPs 2 and 4 with 14.7 million ha and 10.6 million ha, respectively (Fig. 2; Supplementary Table 2). Under SSPs 3 and 5, however, the region is projected to experience a net forest cover loss of 5.2 million ha and 3.1 million ha,

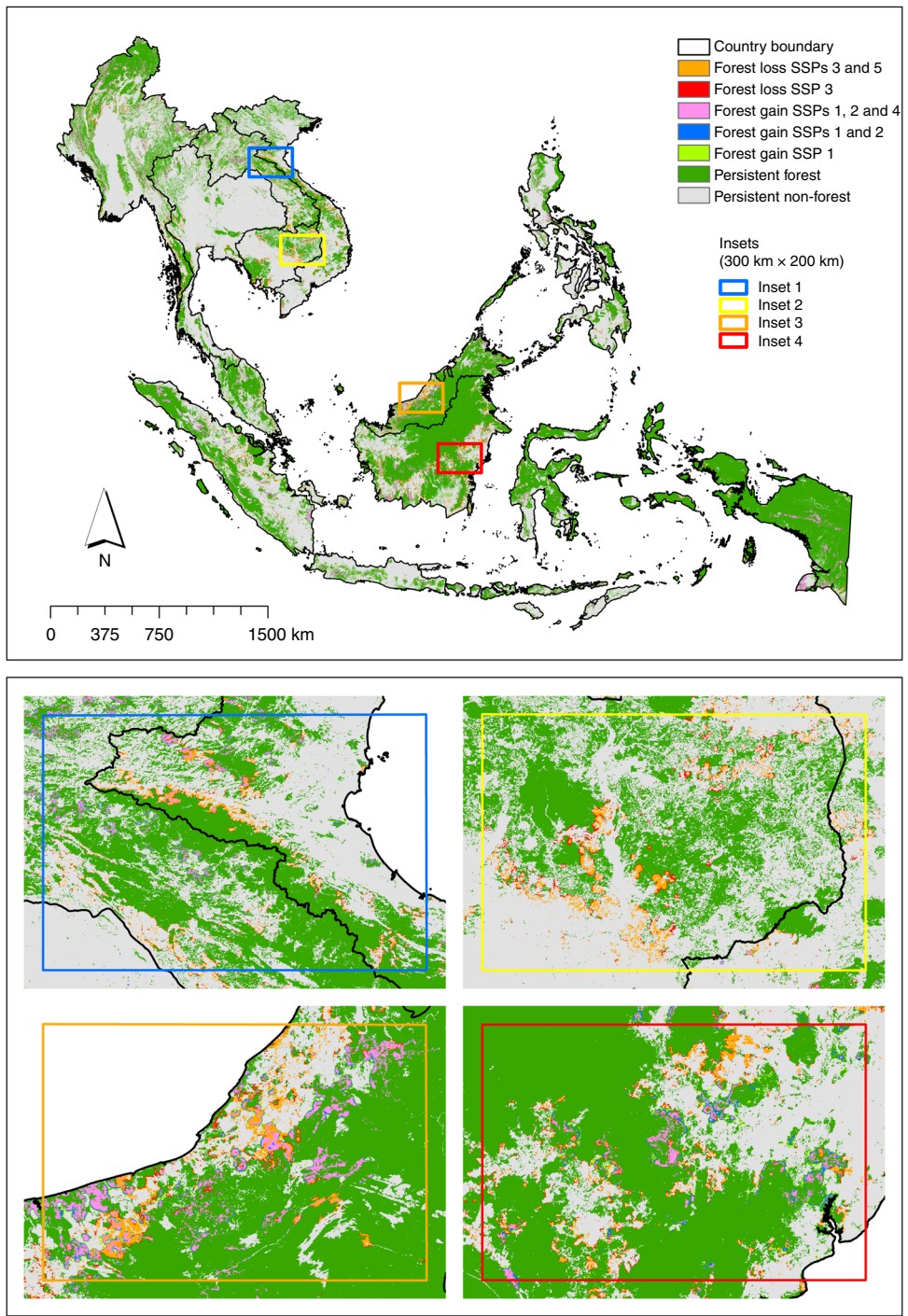

**Fig. 1** Maps showing the spatially allocated projected forest cover changes in Southeast Asia under the five shared socioeconomic pathways (SSPs) (2015–2050). The four insets show the spatially allocated projected forest cover changes in some parts of Laos and Vietnam (inset 1), Cambodia (inset 2), Malaysia (inset 3) and Indonesia (inset 4)

respectively. Under the best-case scenario (i.e. SSP 1), therefore, Southeast Asia would be able to gain a total of 1651 Tg C AFCS, and under the worst-case scenario (i.e. SSP 3), the region would lose a total of 790 Tg C AFCS. Both projected values are 2050 equivalent (see Methods).

At the country level under the best-case scenario, Indonesia would be the highest gainer both in terms of forest cover and AFCS gain, with a 41 and 49% share of the region's total forest cover and AFCS gains, respectively (Fig. 2). The other top gaining countries both in terms of forest cover and AFCS are

Myanmar, Malaysia and the Philippines. Under the worst-case scenario, Indonesia would also be the highest loser, at 48 and 55%, respectively. The other top losing countries are Malaysia, Cambodia, Myanmar and Vietnam. At the province level, in terms of AFCS gain under SSP 1, 57% of the top 30 gaining provinces would be in Indonesia, and 23 and 10% would be in Myanmar and Malaysia, respectively (Fig. 3; Supplementary Table 3). In terms of AFCS loss under the worst-case scenario, majority of the top 30 losing provinces would also be coming from Indonesia (57%), while most of the

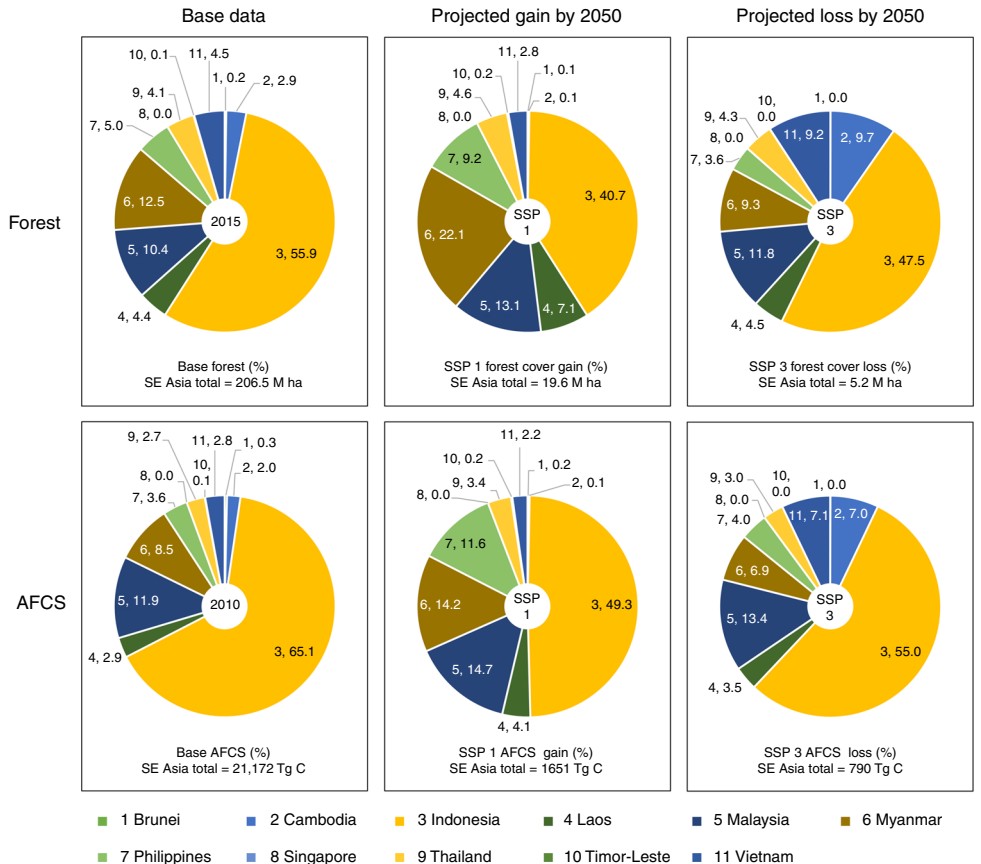

**Fig. 2** Country-level distribution of current forest cover and aboveground forest carbon stock (AFCS) and the projected forest cover and AFCS gains and losses (2015–2050) in Southeast Asia. In each section of the pie charts, the first numerical value refers to the country number, which corresponds to the country number in the figure legend, and the second refers to the percentage share of the country relative to the region's total. Base AFCS is based on the extent of forest cover in 2015. In this figure, only the best-case (SSP 1) and worst-case (SSP 3) scenarios are presented. The data for SSPs 2, 4 and 5, including the complete statistics of forest cover and AFCS changes, are given in Supplementary Table 2

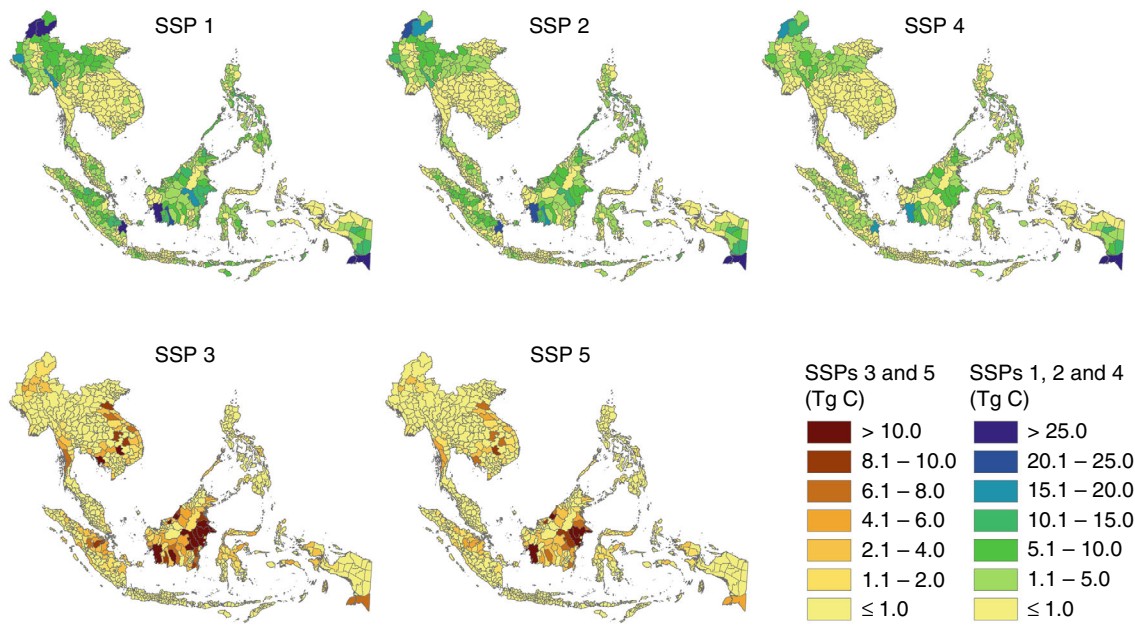

**Fig. 3** Province level distribution of the projected aboveground forest carbon stock (AFCS) gains and losses in Southeast Asia (2015–2050). Under the worst-case scenario (SSP 3), 17 of the top 30 AFCS-losing provinces were found in Indonesia, while the rest were found in Cambodia (3), Malaysia (3), Myanmar (3), Laos (2) and Vietnam (2). Under the best-case scenario (SSP 1), 17 of the top 30 AFCS-gaining provinces were also found in Indonesia, while the rest were found in Myanmar (7), Malaysia (3), Philippines (1), Laos (1) and Thailand (1). The top 30 provinces under each SSP are given in Supplementary Table 3. Note: Indonesia—regency level; Myanmar and Malaysia—district level

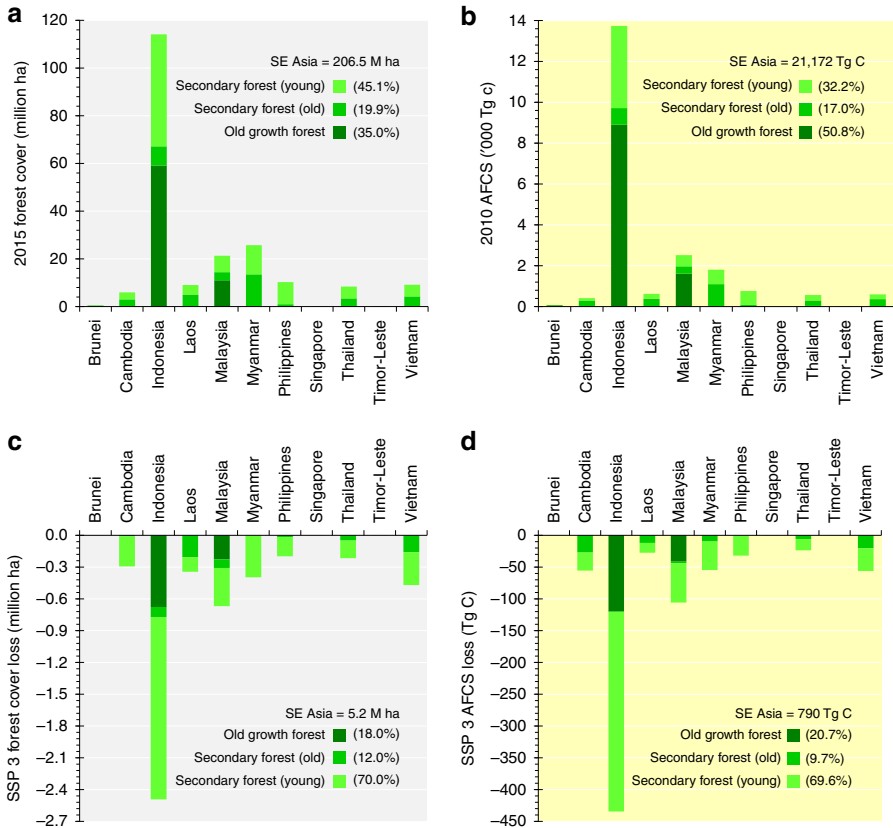

**Fig. 4** Forest cover and aboveground forest carbon stock (AFCS) in Southeast Asia, and their respective losses by 2050 across forest classes under the worst-case scenario (SSP 3). Country-level distribution of the current forest cover and AFCS considering forest classes (**a**, **b**) and country-level distribution of projected forest cover and AFCS losses across forest classes (**c**, **d**). For (**b**), the extent of forest in 2015 was used. The graph for the other forest-losing scenario, SSP 5, is presented in Supplementary Figure 3

rest would be coming from Cambodia (10%), Malaysia (10%) and Myanmar (10%).

**Projected forest and carbon stock losses by forest class.** Our results revealed that 35% of Southeast Asia's forest cover was old growth forest in 2015 (Fig. 4a; Supplementary Figure 1), as per our forest reclassification procedure which considers IFs and aboveground carbon density (ACD) (Supplementary Figure 2). Most of the region's current old growth forest is found in Indonesia (83%) and Malaysia (15%). This forest class currently stores half of the region's current AFCS (Fig. 4b). Based on IPCC's approximate threshold for age and carbon sequestration rates (CSRs) across forest classes and types (Supplementary Table 4; Supplementary Figure 2), 20% of the region's forest is classified as old secondary forest, and the rest is classified as young secondary forest (Fig. 4a). Under SSP 3, 18% of the projected forest cover loss in the region by 2050 would be old growth forest (Fig. 4c), and this would be responsible for 21% of the region's projected AFCS loss by 2050 (Fig. 4d). Under SSP 5, 17% of the projected forest cover loss would be old growth forest, and this would be responsible for 19% of the region's projected AFCS loss (Supplementary Figure 3).

**Projections in intact forests and protected areas.** Currently, Southeast Asia has 38.3 million ha of IFs in total, accounting for about 19% of the region's 2015 total forest cover (Supplementary Figure 4; Supplementary Table 5). Our projections revealed that, by 2050, the region's IFs would be losing forest cover ranging

from 22 thousand ha (SSP 5) to 39 thousand ha (SSP 3), resulting in an AFCS loss of 3 Tg C (SSP 5) to 5 Tg C (SSP 3). Such a loss in AFCS accounts for <1.0% of the projected total AFCS loss under SSPs 3 and 5 (Fig. 2 and Supplementary Table 2). The region is also home to 38.5 million ha of PAs (Supplementary Figure 4; Supplementary Table 5). However, with 362 thousand ha (SSP 5) to 580 thousand ha (SSP 3) of projected forest cover loss, the region's PAs would be losing some fifteen times more forest cover than that in IFs. This forest cover loss would result in an AFCS loss ranging from 44 Tg C (SSP 5) to 71 Tg C (SSP 3), which would account for about 9% of the total AFCS loss under SSPs 3 and 5 (Fig. 2; Supplementary Table 2).

**Model validation.** Our modelled forest transition potential maps (TPMs) (Supplementary Figures 5 and 6) played a key role in the spatial allocation of the projected future forest cover changes. Here, the Skill Measure (SM) statistic was used to assess the predicted power of these TPMs (see Methods). This statistic is a robust validation parameter because it compensates for the dependence of expected accuracy on the number of transitions and persistence classes. In this study, the SM values derived for the 11 countries in Southeast Asia had an overall average of 0.56 (Supplementary Table 6). As this overall average SM value is well above 0 (a case where the model is no better than chance), but more especially above −1 (a case where the model is worse than random chance), our modelled TPMs have the predictive power to correctly simulate the past-to-present spatial patterns of forest cover changes in the region, which are likely to continue into the future. Needless to say, the transition potential modelling was

performed on a large scale, at country level. Generally, our derived SM values are either close to, or within the range reported in other studies[33–35].

## Discussion

Translating the projected changes in AFCS into rates, the forest-gaining scenarios suggest that Southeast Asia would be able to sequester C at a rate ranging from 25 Tg C yr$^{-1}$ (SSP 4) to 47 Tg C yr$^{-1}$ (SSP 1) (Fig. 2; Supplementary Table 2). By contrast, the forest-losing scenarios suggested that the region would be emitting C at a rate ranging from 14 Tg C yr$^{-1}$ (SSP 5) to 23 Tg C yr$^{-1}$ (SSP 3) (Fig. 2; Supplementary Table 2). At a ratio of 1 Mg C to 3.67 Mg $CO_2$[36], the projected changes in AFCS translate to a total projected gain (sink) of 6.1 Pg $CO_2$ under the best-case scenario, SSP 1, and a total projected loss (source) of 2.9 Pg $CO_2$ under the worst-case scenario, SSP 3. To put these projections into perspective, in 2015 alone, Southeast Asia emitted a total of 1.4 Pg $CO_2$ due to fossil fuel use and cement production[37]. This shows that the projected forest cover gain under SSP 1 (Figs. 1 and 2; Supplementary Table 2) would be able to absorb 173 Tg $CO_2$ yr$^{-1}$, about 12% of the above-mentioned 2015 $CO_2$ emissions of the region. On the other hand, the projected forest cover loss under SSP 3 (Figs. 1 and 2; Supplementary Table 2) would result in an emission of 83 Tg $CO_2$ yr$^{-1}$, which is about 6% of the region's 2015 $CO_2$ emissions.

As mentioned earlier, Southeast Asia's IFs and PAs are important reserves for tropical biodiversity and AFCS. However, based on the past-to-present spatial pattern of forest cover changes, our results showed that the region's IFs, as in the recent past[15], would continue to lose forest cover. These projected forest cover losses (Supplementary Table 5) translate to a net loss rate ranging from 636 ha yr$^{-1}$ (SSP 5) to 1121 ha yr$^{-1}$ (SSP 3). While previous studies have shown that PAs would have relatively lower deforestation rates compared to unprotected areas[5,38], they too would continue to lose forest cover, and even at a much higher rate, from 10.3 thousand ha yr$^{-1}$ (SSP 5) to 16.6 thousand ha yr$^{-1}$ (SSP 3), when compared to IFs. A related study reported that one-third of global PAs are under intense human pressure[39] and this magnifies the likelihood of habitat degradation, ultimately leading to loss and extinction of species[40]. Efforts in forest protection and conservation across Southeast Asia's IFs and PAs thus need to be strengthened if the aim is to alter the past-to-present trajectories of forest cover loss. A strong commitment and political will among government leaders toward regular monitoring and evaluation are likewise needed, considering that, according to our analysis, over 88% of Southeast Asia's forests are unprotected, being outside the PAs, and only about 14% of the region's IFs are within the PAs (Supplementary Figure 4; Supplementary Table 5).

Indeed, it is imperative for decision makers to preserve the remaining IFs in the region[17], which represent about 9% of the world's total tropical IFs[13]. IFs are not only better at providing ecosystem services compared to degraded forests, but also help in the restoration of surrounding areas[13,41]. IFs are better at storing carbon than fragmented landscapes[42]. Intact peat swamp forests also have ground water tables that are close to the forest floor, making neighbouring forest biomass permanently moist and resilient to fire[43]. Meanwhile, PAs are important strategies for habitat and biodiversity conservation. Deforestation rates within PAs are significantly lower than outside PAs[5,38], and thus areas where IFs have suffered and/or are currently experiencing habitat loss should be included within the PAs.

Not surprisingly, the projected AFCS gains and losses would not be uniformly distributed across space, that is, across countries and among the provinces within each country. At the country level, Indonesia, Malaysia, Myanmar and the Philippines would be the top nations in terms of future AFCS gain, and Indonesia,

Malaysia, Cambodia, Vietnam and Myanmar would be hotspots for future AFCS loss (Fig. 2; Supplementary Table 2). Our results also revealed the top provinces in terms of AFCS changes across the five SSPs, allowing the identification of hotspot provinces for future forest cover and AFCS losses (Fig. 3; Supplementary Table 3). While old growth forests and IFs need to be protected and preserved[15,44], secondary forests that are outside PAs should also be properly managed. As IFs and PAs gain more attention and protection[5,38], the pressure on secondary forests outside PAs is expected to intensify in the coming years. In fact, our projections showed that almost 80% of the projected future AFCS loss under the worst-case scenario would be from secondary forests (Fig. 4d).

In the SSPs official documentation[28,29,32], land use is strongly regulated under SSP 1, and tropical deforestation rates are greatly reduced, whereas SSP 2 broadly follows historical patterns, with some exceptions and regional disparities in environmental conservation. As such, the countries progress slowly towards the realisation of sustainability goals, and the pressure on tropical forests is reduced due to higher energy efficiencies and mitigation-induced afforestation. Conversely, land use under SSP 3 has limited regulation. Forest mitigation activities are minimal, and deforestation would continue due to competition over land, coupled with the rapid expansion of agriculture. SSP 3, the worst-case scenario, gives little consideration to environmental protection, and so it would result in comparatively high pressure on the global land use system. Under SSP 4, deforestation is also reduced, primarily through international collaboration and the transition to a low-carbon economy. Finally, SSP 5 is characterised by a fossil fuel-intensive economic growth scenario, where a high demand for land for intensive agriculture, coupled with delayed international cooperation, results in deforestation, but at a slower rate compared to SSP 3. Overall, based on these five SSP storylines[28,29,32] and our results (Figs. 1–4; Supplementary Table 2), SSP 1 is the best-case scenario and thus considered to be the most desired pathway. However, one may ask: Is it likely to happen?

The forest transition theory can perhaps help to explain the plausibility of this scenario. The theory suggests that the socio-economic development of a country can prompt the transition from a shrinking to an expanding forest area, as demonstrated in many of the now so-called developed countries where forests contracted in the past when they were still developing[45–47]. Although not framed in the context of this theory, a recent analysis of global land change provides indications that there has indeed been forest expansion in recent decades (from the 1980s to 2010s)[48]. This study found that global tree cover had increased by 7.1% (2.24 million km$^2$) over a 35-year period, owing to the net gain in the extra-tropics outweighing the net loss in the tropics[48].

Some countries in Southeast Asia demonstrated a substantial amount of forest cover gain over the 1992–2015 period, as shown by our results (Supplementary Figure 7), which are consistent with those of other studies[17,22,48]. However, the total forest cover loss still outweighed the total gain. It is therefore necessary to at least slow down the deforestation rate and continue, or even increase the efforts in reforestation and afforestation. Another recent study found that population pressure and food production are important factors driving forest contraction in Southeast Asia, and that agricultural efficiency is among the possible factors affecting forest expansion in the region[22]. As developing nations improve economically, their capacity to provide off-farm employment and improve agricultural productivity may increase, thus leading to the abandonment of marginal agricultural lands[22,47,49]. Forest transition hypothesises that such land

abandonment is to be followed by forest regeneration and tree plantation[46,47,50], thus supporting the likelihood of SSP 1.

Economic development may also strengthen the capacity of developing nations to implement reforestation programmes[51] and/or conserve their own forest resources[52,53]. There is, however, a possible drawback in the latter, relating to the leakage effect, where deforestation may be displaced to neighbouring locations due to the migration of deforestation agents or the trade of timber and agricultural products, thus decreasing the regional and global environmental benefits of policies aimed at conserving natural ecosystems[53]. In addition to socioeconomic development, forest scarcity is another key driver of forest transition[22,54]. The scarcity of forests drives forest conservation and protection aimed at stabilising the market prices of forest products and restoring depleted ecosystem services[54,55]. In Southeast Asia, it has been found that countries experiencing forest expansion since 1990 are also those with relatively less remaining forest area, and that countries with relatively more remaining forest area continue to lose forest cover[22].

The New York Declaration on Forests (http://forestdeclaration.org), if realised, can certainly help improve the likelihood of SSP 1. The declaration is a voluntary and non-binding international initiative that aims to end natural forest cover loss and calls for the restoration of degraded and deforested lands. The New York Declaration's aim also overlaps with the UN's sustainable development goals (SDGs) (https://sustainabledevelopment.un.org), the Paris Agreement (https://unfccc.int/process-and-meetings/the-paris-agreement/the-paris-agreement), the Bonn Challenge (http://www.bonnchallenge.org) and the legally binding Aichi Biodiversity Targets (https://www.cbd.int/sp/targets). Target 11 of the Aichi Biodiversity Targets (2011–2020) states that, by 2020, at least 17% of terrestrial and inland water areas are conserved, a measure that would certainly help in reducing direct deforestation[56]. While Target 11 is currently partly achieved, at 14.8% as of June 2017, it is believed that a similar or more ambitious target would follow in the post-2020 Strategic Plan for Biodiversity[56]. This would, to a great extent, facilitate regional progress in halting deforestation and the better conservation and management of both IFs and PAs.

These international initiatives are important as they, to some extent, help guide our actions and shape our visions for the future. Other international programmes are also available, like the United Nations Programme on Reducing Emissions from Deforestation and Forest Degradation (UN-REDD Programme) which supports developing countries to develop capacities needed to meet the requirements for reducing emissions from deforestation and forest degradation and the role of conservation, sustainable management of forests and enhancement of forest carbon stocks in developing countries (REDD+) (https://www.un-redd.org/; https://redd.unfccc.int/; http://www.fao.org/redd/en/). In addition, the awareness that government leaders and their respective peoples have of various global environmental issues, including deforestation and its widespread consequences, has also undoubtedly increased in the last few decades. Arguably, all of these things can have a significant impact towards forest protection, conservation and expansion, sustaining the likelihood of SSP 1—the sustainability scenario.

It should be stressed, however, that for the projected forest cover gains to be achieved, especially under SSP 1, efforts should focus on both the protection and conservation of the remaining forests and the expansion of forest cover through reforestation and/or afforestation. We recognise the importance of tree plantations for economic purposes in most of the countries in the region, but tree plantations for ecological purposes must also be considered. An example is the Philippines' 2011 forest rehabilitation initiative, called the National Greening Program (NGP),

which initially aimed to plant 1.5 billion trees on 1.5 million ha by 2016, and is a government priority programme to reduce poverty, promote food security, environmental stability and biodiversity conservation, and enhance climate change mitigation and adaptation[57,58]. The programme has been expanded (ENGP; 2016–2028) with the aim of reforesting all remaining unproductive, denuded, and degraded forestlands nationwide, comprising about 7.1 million ha, which also contribute to environment-related risks such as soil erosion, landslides, and flooding[59]. Perhaps, at this stage and in relation to SSP 1, it is important for individual countries to formulate, if not yet available, similar initiatives that also consider the future environmental goal of tree planting or re-greening (i.e. reforestation and/or afforestation for biodiversity and regulating and supporting services), focusing not only on its economic prospect (i.e. tree planting for a near-future harvest of provisioning services). In this regard, strong policy support is needed.

With regards to uncertainties, our results are mostly dependent on the extent of forest cover changes (loss and gain) in recent decades, and that the choice of input data was therefore critical for this study. Without prejudice to the other possible data sources and considering our requirements for data selection (see Methods), we used the ESA-CCI land cover maps as our data source to detect forest cover changes from the recent past to the present. Our derived forest/non-forest (F/NF) maps enabled us to detect both forest cover losses and gains at the same time at the country and province levels (Supplementary Figure 8). Other important databases, such as the Forest Resources Assessment's (FRA) records[60], could not provide such information (i.e. referring to both gains and losses). The FRA's statistics could only provide a net change estimate, but more importantly, do not assess the spatial patterns of forest cover changes and, unlike our study, employ a land use definition of forest that also includes bare areas where trees are expected to regenerate, while excluding areas with tree cover under agricultural or urban land use[61]. In this study, the derived past-to-present forest cover losses and gains were useful in the quantification of the potential future forest cover losses under SSPs 3 and 5 and the potential future forest cover gains under SSPs 1, 2 and 4 in each country (Supplementary Figure 8a). The spatial locations of these past forest cover gains and losses were helpful in the calibration and modelling of TPMs for future forest cover gains and losses (Supplementary Figures 8b, 5 and 6).

Our detected C emission rate of 100 Tg C yr$^{-1}$ in Southeast Asia due to AFCS loss over the 2005–2015 period, through the use of the ESA-CCI land cover-derived F/NF maps, is close to the 118 Tg C yr$^{-1}$ emission rate found by Baccini et al.[14] over the 2004–2014 period (Supplementary Table 1). While Baccini et al.'s emission rate is slightly higher, this is expected because their estimation of C emission was performed on an annual basis, thus also including the increment in biomass due to the growth of trees that occurred before deforestation. Some difference is also related to the discrepancy between the 2000 AGB map that we used and the time-series AGB maps (2004–2014) used by Baccini et al. Nonetheless, this comparative analysis offers confidence in our input data and results.

On the other hand, based on the same input AGB dataset (see Methods), the forest loss data for the 2005–2015 period by Hansen et al.[19] resulted in a C emission rate of 235 Tg C yr$^{-1}$, more than twice as high as our finding based on the ESA-CCI land cover-derived F/NF maps and that of Baccini et al.[14] (Supplementary Table 1). The large difference in quantity between Hansen et al.'s forest loss and our detected forest loss over the 2005–2015 period could explain the apparent large difference in C emission rates. In turn, the difference in forest loss is probably due to the interaction of the different forest definitions and spatial

and temporal resolutions of the two datasets, making it difficult to disentangle the various factors. Certainly, the relatively coarse (300 m) ESA-CCI maps may have missed fine-scale deforestation events which were captured by the high-resolution map (30 m) of Hansen et al. The fine-scale forest losses are mostly typical of continental Southeast Asia, where about 20% of the total forest loss was detected by Hansen et al.'s dataset[62], while the large-scale deforestation events occurring in insular Southeast Asia are considered detectable by the ESA-CCI maps (Supplementary Table 1). That said, the use of fine resolution data is advisable where possible, especially in areas characterised by small-scale forest dynamics.

A global study on greenhouse gas emissions from tropical forest degradation and deforestation between 2005 and 2010 found that 41% (2.56 Gt $CO_2$ $yr^{-1}$) of the emission due to deforestation came from Southeast Asia[16]. At a ratio of 1 Mg C is to 3.67 Mg $CO_2$[36], this translates to an emission rate of 697 Tg C $yr^{-1}$. However, a comparison could not be made, as this estimate includes not only aboveground live biomass, but also other pools, including belowground live biomass, dead organic matter and soil organic matter, which were not considered in our study. This is especially true given that most of the emissions from the peatland forests of Southeast Asia are from the soil carbon pool[63,64].

As in the FRA's records[60], the SSP Public Database, Version 1.1 (https://tntcat.iiasa.ac.at/SspDb) only reports values that would result in a net change estimate when the difference between two projected values at two different time points is determined. As a result, we could only simulate the net change: whether a particular SSP would either be gaining or losing forest cover. A database or study that also reports on both gross forest cover gains and losses[10,17,19] would allow land change models to simultaneously simulate the potential gains and losses of forest, making the forest cover change scenario more realistic. This aspect of the SSP Public Database could thus be considered in its future development.

There is also a need to downscale the SSP's projected socioeconomic variables, such as population and gross domestic product[65–67], to enable researchers to examine how these projected variables would relate to the spatial pattern of future forest cover changes. In such a case, a forest cover change scenario modelling can be conceptualised, and be based purely on the projected quantities and spatial locations of future forest cover changes. The other limitations of our study relate to our land change modelling procedure and our approach in sub-classifying forest. Our land change modelling procedure could not capture the fluctuation (quantity) and direction (loss or gain) of forest cover changes in every 10-year interval within the whole 2015–2050 period, as depicted in the SSP Public Database, Version 1.1. Our approach in sub-classifying the forest class might also have resulted in a conservative result, suggesting that there is much less old growth and old secondary forests (Fig. 4; Supplementary Figure 3). This is because the approach, which is based mainly on IFL, ACD and CSR, did not explicitly take forest degradation into account. For instance, an old growth, or an old secondary forest stand, which has been experiencing forest degradation might have been classified as young secondary forest due to its low ACD. Thus, our results should be interpreted on the basis of the assumptions made (Supplementary Figure 2).

## Methods

### Research design.
In this study, we focused on the spatial allocation of the projected quantities of future forest cover changes in Southeast Asia from 2015 to 2050 under the five baseline SSPs, employing a state-of-the-art spatially explicit land change modelling approach and using remotely sensed data. The potential implications of these spatially allocated projected forest cover changes were examined by quantifying their consequent AFCS changes at the country and province levels, within the IFs and PAs, and across forest classes in Southeast Asia.

### Projected future forest cover data.
The data on projected quantities of future forest cover were downloaded from the SSP Public Database, Version 1.1 (https://tntcat.iiasa.ac.at/SspDb)[28,29,32]. More specifically, the downloaded Microsoft Excel file contained projected areas of various land uses, including forest, from 2005 to 2010, and then at a 10-year interval from 2010 to 2100. For this study, we considered only the forest cover projections from 2015 to 2050 under the five baseline SSPs. The said projected areas of forest cover are limited only to quantities at the global and regional levels and have no spatial dimension. At the regional level, the world is divided into five regions, the OECD, Reforming Economies, Asia, Middle East and Africa, and Latin America and the Caribbean. In the Asian region, SSPs 1, 2 and 4 show an increasing trend in forest cover between 2015 and 2050, and SSPs 3 and 5 show a decreasing trend. SSP 1 would have the highest increase (c. 38 million ha), while SSP 3 would have the greatest decrease (c. 10 million ha) (Supplementary Figure 9).

### Past-to-present forest maps for spatiotemporal modelling.
Past-to-present forest maps were needed in this study to facilitate the analysis. Various global forest and land cover datasets are available, including Hansen et al.'s data (global tree canopy cover in 2000 and c. 2010, forest cover loss between 2000 and 2018 and forest cover gain between 2000 and 2012, all in 30 m spatial resolution) (https://earthenginepartners.appspot.com; https://landcover.usgs.gov)[19], the GLOBE-LAND30 (30 m 2010 land cover map) (www.globallandcover.com)[68], the ALOS PALSAR Forest/Non-Forest maps (25 m forest/non-forest map for 2007–2010, 2015–2016) (www.eorc.jaxa.jp)[69], the Landsat Tree Cover (30 m tree cover continuous field for 2000, 2005, 2010, 2015) (http://glcf.umd.edu)[70], the ESA-CCI (300 m land cover for 1992–2015, annually) (www.esa-landcover-cci.org)[71], and MODIS data (c. 250 m vegetation continuous fields for 2000–2010, annually; c. 500 m land cover for 2001–2012, annually) (http://glcf.umd.edu)[72]. We selected the dataset for this study based on two basic requirements. First, the dataset should be able to cover a span of at least 20 years. And second, the dataset should be spatially and thematically consistent across the years it covers.

The Landsat Tree Cover dataset[70] was a candidate, but a larger time span (from the 1990s) was needed for this study in order to facilitate the detection of forest cover changes, especially forest cover gains, which were needed in the calibration process of the land change model used in this study (Supplementary Figure 8). A decadal time span is often necessary for forest cover gain to become clearly detectable from medium spatial resolution satellite imagery such as those of Landsat. Furthermore, to separate the forest from the non-forest pixels in the Landsat Tree Cover dataset would require an intensive classification procedure via thresholding, as well as rigorous validation of the classification results. A previous study found that the forest classification errors detected for the Landsat Tree Cover dataset, including those of the global tree canopy cover and the MODIS vegetation continuous field, were sensitive to the thresholds used[73].

The ESA-CCI land cover dataset (v2.07), produced by the European Space Agency (ESA)—Climate Change Initiative (CCI)[71], was also among the datasets assessed in the previous study[73]. Focusing on Philippine forests, this study found that the ESA-CCI land cover dataset had the least disagreement with the reference dataset used among the various datasets examined and compared in terms of forest cover[73]. Taking this into consideration, alongside the data selection requirements explained above (i.e. a time span of at least 20 years and spatial and thematic consistency), we decided to use the ESA-CCI land cover dataset. The spatial and thematic consistency requirement ensures that the selected land cover dataset can facilitate forest cover change detection and spatiotemporal analysis.

We downloaded the 1992, 2005 and 2015 ESA-CCI land cover maps from the official website (www.esa-landcover-cci.org) and projected them onto the Asia South Albers Equal Area Conic projection system, hereafter the Albers projection system. The land cover maps have a thematic resolution of 37 land cover categories, which we reclassified into two categories, forest and non-forest, following the reclassification procedure used in Estoque et al.[73]. Forest is therefore defined here as lands classified under any of the following categories as per the ESA-CCI land cover dataset (v2.07): (class codes 50, 60–62) tree cover, broadleaved, evergreen and deciduous, closed to open; (class codes 70–72, 80–82) tree cover, needleleaved, evergreen and deciduous, closed to open; (class code 90) tree cover, mixed leaf type (broadleaved and needleleaved); and (class codes 160, 170) tree cover, flooded, fresh or brackish water, saline water. The resulting maps are herein called forest/non-forest (F/NF) maps.

### Spatiotemporal modelling.
The future projections of forest cover under the five SSPs enabled us to calculate the extents of potential future forest cover changes (losses and gains) between 2015 and 2050 across the countries in Southeast Asia. As mentioned earlier, however, these derived future forest cover changes are limited to quantity and have no spatial dimension. To make these projected future forest cover changes more meaningful and useful, they needed to be spatially allocated.

Studies that are closely related to this present study include those of Kubiszewski et al.[27] and Hasegawa et al.[74]. Kubiszewski et al.[27] spatially allocated the quantities of future global land use/land cover changes projected under the four great transition scenarios at 1 km × 1 km grid resolution. In their spatial allocation of future forest cover change, they applied a rule based on adjacency or neighbourhood. More specifically, under a scenario in which forest area would

increase by 2050, the forest category was set to gain only from its adjacent grids of different land use/land cover categories. Similarly, under a scenario in which the forest category would decrease by 2050, the forest category was set to decrease through its outermost grids. In cases where the total area available was not enough to meet the land change demand, a slack variable and random terrestrial layer were used to absorb gains and losses.

On the other hand, Hasegawa et al.[74]. developed a land use allocation model and applied it in the context of SSP 2. They used carbon stock density in their spatial allocation of projected future global forest cover quantities by 2100 under two scenarios (baseline and mitigation) at $0.5° × 0.5°$ grid resolution. More specifically, afforestation was assumed in non-forest areas with a carbon stock density lower than $2 \text{ kg C m}^{-2}$, a definition used to differentiate forest and non-forest areas in an earlier study[75]. Forest and other natural vegetation were allocated on other lands based on the carbon stock density in each grid but excluding pasture. For the allocated forest area and the statistically determined forest area to be the same, a level of carbon stock density between forest and grassland was determined[74].

In this study, we developed and applied an alternative approach for spatially allocating the projected future forest cover changes (Supplementary Figure 8). Our method was built on a state-of-the-art spatially explicit, pattern-based land change modelling approach[76], and employed the Land Change Modeler (LCM)[77], which is available in a software package called TerrSet. The approach included three major parts: forest cover change quantification, transition potential modelling, and forest cover change spatial allocation.

**Forest cover change quantification**. Using the reclassified F/NF maps, forest cover losses and gains were detected between 1992 and 2015 for the whole of Asia, one of the five SSP regions as described above. The percentage share of each country to the region's past total forest cover loss and gain was determined. These per country percentage shares of forest cover loss and gain were used as a multiplicative factor to proportionally allocate the region's projected forest cover losses and gains (2015–2050) among all the countries in the region under the five SSPs. The entire process resulted in a projected quantity of forest cover loss or gain per country under each SSP (Supplementary Figure 8a). Here, we only focused on Southeast Asia, and thus, we only used the results for the 11 Southeast Asian countries. The country boundary layer was sourced from https://gadm.org.

**Transition potential modelling**. The overall purpose of our transition potential modelling was to locate or identify the areas or pixels with higher propensity or likelihood to gain (under SSPs 1, 2 and 4) and lose (under SSPs 3 and 5) forest, so that the projected quantities of forest cover loss and gain per country in Southeast Asia (2015–2050) can be spatially allocated. To do this, we used LCM's Multi-Layer Perceptron Neural Network (MLP NN), which employs the back-propagation algorithm (see also TerrSet Help System[78]). Country by country, the F/NF maps for 2005 and 2015, together with seven spatial driver variables, were used as input into the transition potential modelling framework in which MLP NN was used. The seven spatial driver variables included elevation, slope, distance to road, distance to urban area, forest cover loss or gain share per province, distance to deforested or reforested area, and distance to forest edge, inward or outward (Supplementary Figure 8b). The elevation map, resampled (bilinear) to 300 m, was derived from the 30-m Shuttle Radar Topography Mission (SRTM) data (www2.jpl.nasa.gov/srtm/). The slope map was derived from the elevation map. The distance to road map was produced based on the road network map available from www.diva-gis.org. Distance refers to Euclidean distance for all the distance variable maps. The province boundary layer was sourced from https://gadm.org. Like the derived F/NF maps, all the spatial data were projected onto the Albers projection system.

The transition potential modelling (Supplementary Figure 8) included the processes of training and testing using sample pixels that were randomly selected by the LCM's MLP NN (Supplementary Table 7). For instance, for the transition forest to non-forest, samples of equal size were randomly selected from both the pixels that experienced forest loss and those pixels that were eligible to change but did not (persistence). Of the total samples selected, half (50%) was used for training, while the other half (50%) was used for testing. Based on the training samples and the resulting transition potentials of the pixels, the model performed a F/NF classification, the results of which were compared against the testing samples (see Model Validation section below). The pixel values of the TPMs (Supplementary Figures 5 and 6) were considered probability of change given that the prior probability of change was 0.5 because of the equal size of samples from the changed pixels and non-changed pixels (persistence). More details can be found in TerrSet Help System[78].

**Forest cover change spatial allocation**. Two types of input were needed in our country level forest cover change simulation per SSP, and these were the quantified forest cover change (loss or gain) and the TPM for forest cover change (loss or gain) (Supplementary Figure 8c). The LCM allows the user to input a transition matrix. In our case, this was a $2 × 2$ transition matrix that contained the quantities of forest and non-forest pixels that would change and persist at the end of the simulation period (2050). These quantities were expressed as proportions of the total number of pixels in the forest and non-forest categories at the start of the

simulation period (2015) (see Supplementary Table 8 for details). The model then allocated the quantity of change based on the TPM, first selecting the pixels with the highest probability to change until the change demand was met. We assumed that forest would not replace the existing pixels of built-up (urban areas) and water bodies. The results of our simulations were F/NF maps per country per SSP by 2050, which we later mosaicked. Forest cover change maps (2015–2050) were produced for the whole of Southeast Asia and used in the subsequent analyses.

**Model validation**. The MLP NN algorithm in LCM outputs a statistic called SM which indicates the predicted power of the TPMs (Supplementary Figures 5 and 6). SM is a robust validation measure because it compensates for the dependence of expected accuracy on the number of transitions and persistence classes. It is calculated as Eq. (1)[78]:

$$\text{SM} = \frac{A - E(A)}{1 - E(A)} \quad (1)$$

where $A$ is the measured accuracy based on a confusion matrix. $E(A)$ is the expected accuracy, expressed as: $E(A) = 1/(T + P)$, where $T$ is the number of transitions in the model and $P$ is the number of persistence classes or the number of *from* in the sub-model. SM has a value ranging from −1 (worse than chance) to +1 (perfect prediction), where zero indicates no better than chance. Our model validation results are given in Supplementary Table 5.

**Detecting losses in carbon stock from 2005 to 2015**. We detected the AFCS loss in Southeast Asia between 2005 and 2015 based on the forest loss detected from our derived F/NF maps and an aboveground biomass (AGB) dataset for the year 2000, which we sourced from http://data.globalforestwatch.org. The 2000 AGB represents aboveground live woody biomass density and has a spatial resolution of ~30 m, with data values expressed in $\text{Mg ha}^{-1}$. We first projected this dataset to the same projection we used for the ESA-CCI land cover dataset before aggregating (mean) it to 300 m spatial resolution to be consistent with our F/NF maps. The carbon (C) content of the biomass data was then determined using a ratio of 0.5[79]. The pixel values of the resulting aboveground C density (ACD) map were expressed as $\text{Mg C ha}^{-1}$. We then masked this ACD map with the detected forest cover loss before proceeding with the calculation of the total AFCS loss in each country, $Y$, between 2005 and 2015 (Eq. (2)):

$$\text{AFCS Loss}_{Y(2005-2015)} = \sum_{i=1}^{n} \text{ACD}_{i,2000} × A_i \quad (2)$$

where $\text{ACD}_i$ and $A_i$ refer to the 2000 aboveground carbon density ($\text{Mg C ha}^{-1}$) and the land area (ha), respectively, of pixel $i$; and $n$ is the number of pixels of the detected forest cover loss (2005–2015) with an ACD value within each country, $Y$.

For purposes of comparison, we repeated the above procedure using a different forest loss dataset over the same period (2005–2015), this time from Hansen et al.[19]. To maintain consistency in procedure and facilitate geoprocessing, we also projected Hansen et al.'s 30 m forest loss data to the same projection we used for the other datasets before aggregating (majority) them to a 300 m spatial resolution. We also compared our results on AFCS loss per country with the results from Baccini et al.[14].

**Projecting changes in carbon stock from 2015 to 2050**. To estimate the potential future AFCS loss under SSPs 3 and 5 due to forest cover loss by 2050, we first sub-classified the forest class into old growth forest, old secondary forest and young secondary forest based on the spatial coverage of the 2016 intact forest landscape (IFL)[80], the 2010 ACD data and CSRs (see Supplementary Figure 2; Supplementary Table 4). This is because primary or old growth forests and secondary or second growth forests differ in their respective C stocks and CSRs. Primary forests store at least 30% more C than secondary forests (including degraded and logged forests), primarily because most living biomass C is contained in large, old trees[80,81]. In the Moluccas, Indonesia, for instance, the AGB of primary forests was 2.5 times higher than that of secondary forests in the area[80]. On the other hand, the CSRs of secondary forests are higher than those of old growth forests (Supplementary Table 4). Secondary forests even differ in CSRs between themselves. For instance, old secondary forests have lower CSRs than young secondary forests[82]. CSRs are also dependent on geographic location, for example, continental vs. insular[82]. Supplementary Figure 2 outlines our approach in sub-classifying the forest class into old growth forest, old secondary forest and young secondary forest and its annual sub-classes, and Supplementary Table 4 provides the CSRs we used, considering forest classes and types and geographic location.

The three most important input data sources in our approach for forest sub-classification were the 2015 F/NF map, the 2016 IFL dataset[80] and the 2010 AGB dataset[83], converted into ACD (Supplementary Figure 2). The 2016 IFL dataset is in the form of a polygon layer. The 2010 AGB dataset has a spatial resolution of 100 m and it is the most recent publicly available dataset at 100 m spatial resolution. A 2015 AGB dataset, coinciding with the year of our current F/NF map, would have been ideal, but such data are not available. To our knowledge, the most recent pantropical ACD dataset (2014) with an approximate spatial resolution of 500 m was produced by Baccini et al.[14]. Unfortunately, the 2014 ACD raster layer is not publicly available (www.thecarbonsource.org). We thus used the 2010 AGB

dataset from Santoro et al.[83] in this study. The results of Baccini et al.[14] have, however, been considered in our comparative analysis, as described above. The other datasets used included the polygon layers of the 2010 ecological zones (ecozones)[84] and the continental and insular geographic areas[85]. Like the other datasets, all these datasets were also projected onto the Albers projection system. The 2010 AGB dataset, with data values expressed in Mg ha$^{-1}$, were also aggregated (mean) to 300 m spatial resolution. The C content of this biomass data was also determined by using a ratio of 0.5[79] and values were also expressed in Mg C ha$^{-1}$ (ACD).

After the sub-classification of the forest class, we projected the future (2050) ACD of all the 2015 forest pixels with corresponding 2010 ACD data, and those that were within the spatial extents of the ecozones and geographic location data layers. Our calculations considered the CSRs across forest classes, types, sub-classes and geographic location (Supplementary Table 4). Eq. (3) was used for old growth forest and old secondary forest.

$$\text{ACD}_{js,2050} = \text{ACD}_{js} + (\text{CSR}_{js} \times \text{GP}) \tag{3}$$

where ACD$_{js}$ and CSR$_{js}$ refer to the 2010 above ground carbon density (Mg C ha$^{-1}$) and carbon sequestration rate (Mg C ha$^{-1}$ year$^{-1}$), respectively, of pixel $j$ which is a member of sub-class $s$ of old growth forest and old secondary forest; and GP refers to the growth period (40 years) for old growth forest and old secondary forest from 2010 (the year of the ACD) to 2050 (the end year of the projection). Examples of old growth forest and old secondary forest sub-classes are Old Growth Forest-Tropical Rainforest-Insular and Old Secondary Forest-Tropical Dry Forest-Continental, respectively (Supplementary Table 4).

The ACD values of the sub-classes of young secondary forest (i.e. age ≤ 20 years) were first projected to old secondary forest-equivalents (i.e. 21-year-old equivalent) using their respective CSRs and based on their respective ages (Supplementary Table 4) (Eq. (4)), where their respective ages were obtained by dividing their ACD by their respective CSRs (Supplementary Figure 2).

$$\text{ACD}_{ks,21yo} = \text{ACD}_{ks} + (\text{CSR}_{ks} \times (T - G_{ks})) \tag{4}$$

where ACD$_{ks}$, CSR$_{ks}$ and $G_{ks}$ refer to the 2010 above ground carbon density (Mg C ha$^{-1}$), carbon sequestration rate (Mg C ha$^{-1}$ year$^{-1}$) and age, respectively, of pixel $k$ which is a member of sub-class $s$ of young secondary forest; and $T$ is the minimum age for old secondary forest (21 years). An example of a young secondary forest sub-class is Young Secondary Forest-Tropical Rainforest-Insular-Age 1 (Supplementary Table 4).

The ACD projection to 2050 was then performed using Eq. (3), with GP becoming (GP = $(2050 - 2010) - (T - G_{ks})$), where 2010 is the year of the ACD and 2050 is the end year of the projection. This process considers the transitioning of young secondary forest into old secondary forest during the AFCS projection period (2010–2050), requiring the use of different CSRs. The transition of old secondary forest into old growth forest, however, could not be considered because of the lack of a threshold for age that could separate these two forest classes (Supplementary Table 4).

After projecting the 2010 ACD to the year 2050, the forest classes, types and their sub-classes were all mosaicked. The pixels that corresponded to the projected forest losses from 2015 to 2050 under SSPs 3 and 5 (Fig. 1) were then extracted from the mosaicked 2050 ACD map, and the total potential future AFCS loss in each country, $Y$, and province, $V$, was calculated (Eq. (5)):

$$\text{AFCS Loss}_{Y,V(2015-2050)} = \sum_{i=1}^{n} \text{ACD}_{i,2050} \times A_i \tag{5}$$

where ACD$_i$ and $A_i$ refer to the 2050 projected above ground carbon density (Mg C ha$^{-1}$) and the land area (ha), respectively, of pixel $i$; and $n$ is the number of pixels of the projected forest cover loss (2015–2050) with an ACD value within each country, $Y$, or province, $V$. Here, it is assumed that deforestation will happen in 2050.

To estimate the potential future AFCS gain due to forest cover gain under SSPs 1, 2 and 4, we assumed that trees would be planted not at one time, but rather across the 35-year period (2015–2050). Thus, to estimate the potential future AFCS gain, we used 17.5 years as the GP for afforestation/reforestation, along with the CSRs for young secondary forest (Supplementary Table 4). The projected forest cover gains were first sub-classified according to ecozones and geographic locations to match with the CSRs, followed by the calculation of the 2050 ACD values (Eq. (6)):

$$\text{ACD}_{qs,2050} = \text{CSR}_{qs} \times \text{GP} \tag{6}$$

where CSR$_{qs}$ refers to the carbon sequestration rate (Mg C ha$^{-1}$ year$^{-1}$) of pixel $q$ of the projected forest cover gain which is a member of sub-class $s$ of the new young secondary forest; and GP refers to the growth period (17.5 years) for the projected young secondary forest.

After calculation, the new young secondary forest sub-classes, now with 2050 ACD values, were mosaicked and the total potential future AFCS gain by each country, $Y$, and province, $V$, under SSPs 1, 2 and 4 was calculated (Eq. (7)):

$$\text{AFCS Gain}_{Y,V(2015-2050)} = \sum_{q=1}^{m} \text{ACD}_{q,2050} \times A_q \tag{7}$$

where ACD$_q$ and $A_q$ refer to the 2050 projected aboveground carbon density (Mg C ha$^{-1}$) and $A_q$ is the land area (ha), respectively, of pixel $q$; and $m$ is the number of pixels of the projected forest cover gain (2015–2050) within each country, $Y$, or province, $V$.

**Carbon stock losses in intact forests and protected areas**. Following Eq. (5), we summarised the projected future AFCS losses, including the projected forest cover losses themselves, across forest classes (old growth forest, old secondary forest and young secondary forest) and the other landscape types of high ecological importance, the IFs and PAs. The IFs dataset was the 2016 IFL dataset[15] (www.intactforests.org), which had been used earlier (Supplementary Figure 2). The PAs dataset was the World Database on Protected Areas (WDPA) dataset[86] (www.protectedplanet.net). Like the IFL dataset, the WDPA dataset comes in the form of a polygon layer, which we also projected onto the Albers projection system.

**Reporting summary**. Further information on experimental design is available in the Nature Research Reporting Summary linked to this article.

## Data availability
The sources of all the data used are given in the Methods section. The raster dataset for the simulated future forest cover losses and gains across the five baseline SSPs are available at https://doi.org/10.5281/zenodo.8162177.

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

## Acknowledgements

This work was supported by the Ministry of Environment, Japan, through Research Grant S2–1708, and the Japan Society for the Promotion of Science (JSPS), through Grant-in-Aid for Scientific Research (KAKENHI) 18H00763. R.C.E., M.O., Y.H. and T.T. were supported by the Climate Change Adaptation Program of the National Institute for Environmental Studies (NIES), Japan. The conclusions and recommendations presented in this article are of the authors and do not, in any way, represent the views of the funders or the authors' respective institutions.

## Author contributions

R.C.E. conceived and designed the study, conducted the research and wrote the paper. M.O. and Y.H. provided research supervision and guidance to R.C.E. and helped in the interpretation and analysis. V.A. helped in the data collection and processing, interpretation and analysis and writing. R.D. helped in the interpretation and analysis, and writing. T.T. and Y.M. helped in the interpretation and analysis. All authors approved the manuscript.

## Additional information

**Competing interests:** The authors declare no competing interests.

