## [Peer Review File · Nature Communications]

Reviewers' comments:

Reviewer #1 (Remarks to the Author):

I would like to point out that this is a very interesting study, which would be of great interest for the whole community. However, my major concern deals with the question whether the ESA-CCI input data is suitable for detecting also smaller-scale deforestation events (see below for more detail) and thus allow properly modelling future (especially nation-scale) scenarios. This concern is underpinned by the more than 51-fold discrepancy between this study's carbon emission result and another important study (Pearson et al 2017; mentioned in this manuscript) that uses spatially higher resolution activity data. The authors should at least address this (and some further – see below) concerns before publication. However, having said so, I would like to encourage the authors to follow below suggestions and resubmit this generally very interesting work!

First, however, I must state that I am not an expert on modelling or model validation. In my opinion, SM, SSP and RCP and their results (is a SM of 0.56 sufficiently good?) are not self-explaining for non-experts. Therefore, a short explanation would be helpful – especially for readers without in-depth modelling background. So, regarding the modelling part of the study, I am not able to comment properly on this. Therefore, I would recommend that at least a further reviewer opinion is required for a better assessment of that side of the study.

My following comments will therefore focus only on the remote sensing-based input data as well as the whole forest-related situation (status of deforestation) within the SE-Asian realm. Please find below my major concerns that should be properly addressed before a publication:

- The authors state an enormous difference (>5000% !!!) between carbon emissions of this study (13.5 Tg C/year) and the Pearson study (697.3 Tg C/year). This cannot be explained only by the inclusion of other carbon pools. The authors therefore mention further possible explanations such as the use of different activity datasets (ESA-CCI against Global Forest Cover data by Hansen) and different biomass maps. However, what does this implicate? How reliable is the result of this study?

As a reviewer, I wonder whether the ESA-CCI datasets (1000m and 300m spatial resolution) are good enough to sufficiently depict not only large-scale deforestation patterns but also smaller-scale changes that might have been better depicted by the 30m Global Forest Cover data (and thus result in much lower carbon estimates as compared to a study using the latter dataset). In case the ESA-CCI maps might miss actual (smaller scale) deforestation events, this will certainly affect the modeled deforestation values for 2050 and might be a very likely explanation for this enormous difference between both studies.

Therefore, it might be advisable and very interesting to run the model (even for shorter periods) on the Global Forest Cover data by Hansen (e.g. 2000-2015) as well and compare and discuss their results with each other.

- Forest/forest change patterns on maps (Fig. 1 and supplementary Fig. 1) differ from own experiences and observations of satellite imagery substantially. Example 1 Cambodia (Fig. 1a and supplementary Fig. 1): too little deforestation in the Boeng Peae Wildlife Sanctuary in the 2015 situation and therefore basically no modeled forest loss in that area for 2050 (which is by now in 2018 already to a large extent deforested). Example 2 Borneo (Fig. 1c and supplementary Fig. 1): main modeled deforestation occurs in areas in the north of Sarawak along the coast where there seems to be not much forest left in 2015 (maybe map scale too low). However, along the southeastern border of the forest there is even under the worst scenario SSP 3 no further forest loss expected, even though there was quite a substantial amount of deforestation detected in 2015.

This as well could be an indication that the used input data (ESA-CCI maps) might be not appropriate to reliably depict smaller-scale deforestation patterns.

- When looking in detail at Fig. 1, the modeled map does not indicate any change in several countries where there is – according to known trends and statistics – substantial change ongoing (e.g. deforestation statistics in Myanmar and Laos, see FAO FRA).

Besides above major concerns I have some further points that need clarification:

- The analysis of the ESA-CCI maps as input for their models let the reader deduce that the same forest definition (as for the ESA-CCI maps) must have been applied in this study. However, it would be good if the authors could clearly describe their applied forest definition as it is not explicitly mentioned in their manuscript.

- Regarding the modeled gains in forest cover, it would be very interesting to know for some example countries what could cause that gain under SSP 1 (e.g. forest plantations, abandoned areas, etc.)? I personally would like to know that in the case of Malaysia as I cannot imagine where and how this could take place (is it realistic that plantation areas will be abandoned – especially with the expected elevated resource-demands of a growing human population)?

- In this context I would like to ask the authors to resize both figures 1 (of the main manuscript as well as in of the supplementary data) and the sub-figures to be split into separate figures at reasonable scale to be able to allow the reader to evaluate better the derived maps.

- In the manuscript it would be better if the authors could clearly explain (already in the results section) why they also analyzed IF and PA areas? Right now, these results are described next to the total forest figures without any explanation, leading most probably to some confusion of the reader.

- At least in the supplementary data the authors could consider adding a map of IF and PA at reasonable scale. This could be very informative for the reader.

- In this context, the two results sections (lines 101-142) describe in detail the various percentage values of forest cover gain or loss per country. This should better be condensed in one or two tables.

- Within the discussion section (Comparison, Uncertainties and Insights), results of ESA-CCI figures and FRA FAO records are compared to each other. Such kind of direct comparison is, however, not directly feasible because the former show changes in 'forest cover', while the latter elaborate on 'forest land' ('land use'), which is not the same.

Reviewer #2 (Remarks to the Author):

As my expertise is specifically in land change modeling, and not in scenarios of forest change and their implications, my comments are focused almost exclusively on the land change modeling process. The authors are to be commended on an effective use of the technology deployed, and the model skill exhibited is good.

On lines 424-426, they state "Using information from training samples, MLP NN performs a non-parametric regression analysis between input variables and one dependent variable with the output containing one output neuron, i.e. the predicted memberships". This is not strictly correct. MLP actually has two output neurons for each transition modeled. For example, if the transition being modeled is forest to non-forest, a sample is taken of pixels that experienced that transition

during the historical period (change), and an equal size sample of pixels that were eligible to change, but which did not (persistence). In the implementation in LCM, the user doesn't see the persistence class, but it is there. As a result, the procedure used is not a regression, but rather, a classification. The transition potential that the user sees is the activation level of the output neuron of change. This is indeed a probability (see lines 427-428), but a special one – it is the probability of change IF the prior probability of change were 0.5. It has this prior probability because the system used a balanced sample (equal samples of change and persistence). Because of this special character, transition potentials are only used in a relative sense when allocating predicted change.

The sentence on lines 429-431 is effectively true, but be aware that behind the scenes, there were also transition potentials generated for the two persistence classes (which were discarded, since they were unnecessary for the allocation stage of the prediction).

Line 476/477 is confusing and could benefit from some clarification. Mercator is not an equal area projection. How then were areas calculated?

I hope this helps.

Ron Eastman

Reviewer #3 (Remarks to the Author):

This paper describes an effort to spatially explicitly simulate forest cover transitions in South East Asia under two scenarios. The authors used readily available remote sensing data to detect forest cover loss and gain, seven static spatial determinants to calculate transition probability and a land change model to simulate the changes up to 2050. In addition, they link their forest cover gain and loss with aboveground forest carbon stock. Overall, the paper is written in fluent English, the context of the approach is well defined and the topic is important and merits to receive attention. My main concerns are with the rationale of the methodological strategy and are appended below: I agree with the authors that spatial allocation and mapping of projected large-scale quantities of forest cover change are important for many reasons. What puzzled me (and potentially will be puzzling to the readers) is that, on the one hand, in the Asian region, increasing trends are shown in SSPs 1, 2, 4 and decreasing trends in SSPs 3 and 5 (lines 347-349). On the other hand, they choose to simulate and consider, as the most plausible, the two scenarios with the opposing ends of the spectrum (line 75). If the authors performed the analysis for all five scenarios and they chose to present only the two contrasting, I would advise presenting the full range of spectrum as I believe the paper will benefit greatly (and is much more aligned with the title of this paper). Not only it would capture the full range of plausibility but also the potential implications of each pathway would be better pronounced.

Specific comments

Line 67: is there any measurement of plausibility for Asia? How these two extremes are equally plausible?

Line 94: What does the overall average of 0.56 mean?

Figure 1 needs some improvements e.g. countries name to help the reader navigate and understand the geographical context.

Figure 2 needs to be provided in better resolution.

Lines 416-431/Transition potential modeling: The authors used 2005-2015 changes to train a Neural Network model that in turn created the transition potential surfaces. I believe their methodology suffers from two limitations that have a significant effect on the results.

First, normally in land change modeling and especially in the transition potential modeling part, it is feasible to detect changes for a period of time and incorporate in the model several aspects of these changes using environmental and socio-economic determinants that explain these changes.

The most meaningful way to do so, is to detect changes eg for this case 1992-2005, use as many determinants useful to explain these changes, create the transition potential surface, simulate up to 2015 and compare the simulation product vs the reality (reclassified map of 2015). This calibration process must be repeated until a level of accuracy is reached and is crucial the performance capability of a model.

Second, and equally important, by using actual 1992-2015 forest cover loss as a determinant you introduce to your model a large amount of bias.

Lines 422-423: It is not clear what procedures were followed per scenario. Given that two contradicting scenarios were chosen, this paper only uses static variables to spatially determine a complexity. How the driver variables differ per scenario? I suspect that population, economics and other proxies are important determinants as well and should vary accordingly per scenario (as the authors also mention in lines 218-230).

Line 424: How many samples, per country or province. Also, it is important to mention the sampling strategy (eg random, stratified, weighted in changed areas etc).

Lines 443-452/Model Validation: This part is not clear as it currently stands. Which are the inputs for calculating SM? I suspect that this step is affected.

Detailed Response to Review Comments and Suggestions

Editor's Comments

Your manuscript entitled "Southeast Asia's Forest Futures, 2015-2050" has now been seen by 3 referees, whose comments are appended below. You will see from their comments copied below that the referees acknowledge the potential interest and importance of your work, but between them, they also raise a number of concerns, which must prevent us from offering to publish the paper in its present form. It is not clear to us whether you will be able to address all the concerns raised. If you would like to pursue publication in Nature Communications, we will therefore need to see your responses to the criticisms raised and to some editorial concerns, along with a revised manuscript, before we can reach a decision regarding publication.

RESPONSE: First of all, we sincerely thank the Editorial Office for considering our paper and sending it out for peer review. Considering the comments/suggestions from the reviewers, we have carefully revised our manuscript. Yes, we would like to pursue publication in Nature Communications.

The referees' reports seem to be quite clear. Naturally, we will need you to address all of the points raised. Specifically, for publication in Nature Communications to be appropriate, you must provide compelling support for your selected ESA-CCI input data (Reviewer #1). You will also need to provide explanation and clarifications regarding modelling process (Reviewer #1) and statements (Reviewer #2). Aside from the required clarifications, we will also need you to justify the methodological strategy (Reviewer #3).

RESPONSE: Thank you very much for summarizing the points raised by the reviewers. Our point-by-point responses are given under each Reviewer's comments/suggestions.

In addition, we have summarized the major revisions we have made, as follows:

- 1) To address the issue on input data (Reviewer #1), we have revised and improved our estimates using a new biomass (AGB) map, and thoroughly assessed the impact of alternative datasets on the emission estimates. The updated results reconcile and explain the variability in the historical emission estimates. In particular, we have used the latest global 2010 AGB dataset (100 m) available, released just recently (Santoro et al. 2018) as basis for the emission estimates until 2050. Secondly, we also estimated the emissions using the Hansen et al. (2013) 30 m forest loss data (2005-2015), as suggested by Reviewer #1, along with a global 2000 AGB dataset (30m), and compared these results with the estimates based on the forest loss derived from the ESA-CCI land cover maps along with the same 2000 AGB map. We also compared the results from these two assessments with the findings of Baccini et al. (2017) for Southeast Asia.
- 2) To address the issue on projection (Reviewer #2), we have re-run all simulations and re-computed all indices.
- 3) For our methodological strategy, we basically agree to the points raised by Reviewer #3. However, we also believe that our approach is methodologically sound and acceptable. Please find below our clarifications and justifications.
- 4) We have also improved our approach for projecting aboveground forest carbon stock (AFCS) by considering ecological zones, geographic location and age-based carbon sequestration rates.
- 5) All figures and supplementary materials were also improved.

Reviewer #1 (Remarks to the Author):

I would like to point out that this is a very interesting study, which would be of great interest for the whole community. However, my major concern deals with the question whether the ESA-CCI input data is suitable for detecting also smaller-scale deforestation events (see below for more detail) and thus allow properly modelling future (especially nation-scale) scenarios. This concern is underpinned by the more than 51-fold discrepancy between this study's carbon emission result and another important study (Pearson et al 2017; mentioned in this manuscript) that uses spatially higher resolution activity data. The authors should at least address this (and some further – see below) concerns before publication. However, having said so, I would like to encourage the authors to follow below suggestions and resubmit this generally very interesting work!

RESPONSE: We sincerely thank Reviewer #1 for the time and effort in reviewing our manuscript. Also, for the compliments and encouragement. We really appreciate it. We have considered all the comments and suggestions. Below is our point-by-point response.

First, however, I must state that I am not an expert on modelling or model validation. In my opinion, SM, SSP and RCP and their results (is a SM of 0.56 sufficiently good?) are not self-explaining for non-experts. Therefore, a short explanation would be helpful – especially for readers without in-depth modelling background. So, regarding the modelling part of the study, I am not able to comment properly on this. Therefore, I would recommend that at least a further reviewer opinion is required for a better assessment of that side of the study.

RESPONSE: Thank you very much for the comment, advice and candidness. For the issue on SM, we acknowledge that it would be difficult to categorically claim that 0.56 is sufficiently good. It is because any SM value that is above 0 indicates that the model is better than random chance. Like correlation coefficient, r , generally, the closer the SM to 1, the better. But there are also cases where a 'low' correlation would be enough, and a 'high' correlation would still be not enough, depending on the subject being investigated. This is also true to SM. In our case, we believe 0.56 is good enough because our simulation is performed at the country level. In a sub-national level study (e.g. watershed level, province or city level), 0.56 may still be viewed as good but inferior. Nevertheless, we have clearly defined the value range of SM for readers to understand what it is all about (please see Lines 149-161; 522-531 of the revised manuscript).

My following comments will therefore focus only on the remote sensing-based input data as well as the whole forest-related situation (status of deforestation) within the SE-Asian realm. Please find below my major concerns that should be properly addressed before a publication:

- The authors state an enormous difference (>5000% !!!) between carbon emissions of this study (13.5 Tg C/year) and the Pearson study (697.3 Tg C/year). This cannot be explained only by the inclusion of other carbon pools. The authors therefore mention further possible explanations such as the use of different activity datasets (ESA-CCI against Global Forest Cover data by Hansen) and different biomass maps. However, what does this implicate? How reliable is the result of this study? As a reviewer, I wonder whether the ESA-CCI datasets (1000m and 300m spatial resolution) are good enough to sufficiently depict not only large-scale deforestation patterns but also smaller-scale changes that might have been better depicted by the 30m Global Forest Cover data (and thus result in much lower carbon estimates as compared to a study using the latter dataset). In case the ESA-CCI maps might miss actual (smaller scale) deforestation events, this will certainly affect the modeled deforestation values for 2050 and might be a very likely explanation for this enormous difference between both studies. Therefore, it

might be advisable and very interesting to run the model (even for shorter periods) on the Global Forest Cover data by Hansen (e.g. 2000-2015) as well and compare and discuss their results with each other.

RESPONSE: Thank you very much for the comments and suggestions. We have carefully thought about this issue. We improved the input data, revised our methodology and related estimates, evaluated the Global Forest Cover dataset by Hansen et al., and included new comparative studies for the historical period (before 2015). These changes reduced greatly the difference between our results and other published estimates on historical emissions, which are now of the same order of magnitude. In particular, we addressed and discussed the use of the Hansen dataset in SE Asia. We acknowledged the advantage of fine-resolution data but also consider that this would not change dramatically our results, if most of the emissions are due to large-scale events rather than small-scale changes as suggested by the literature. Moreover, we better understood and explained the large difference with the Pearson et al. study. Here is a more detailed description of our revision:

- (1) We realized that comparing our results for future projection (2015-2050) with the results of Pearson et al. for 2005-2010 is not appropriate. First, it is because the time periods are not the same and that annual deforestation rates may not be the same also (annual deforestation rate from Pearson et al. is not mentioned). Second, and possibly the most important reason, Pearson et al.'s estimates included C pools other than aboveground C stock, especially soil carbon that is responsible of the largest part of emissions in peatland forests.
- (2) To further address this issue and to provide support and indication of reliability on our results, we compared them with the more appropriate (we believe) datasets.
 - (2.1) First, we used a 2000 30m AGB dataset (<http://data.globalforestwatch.org>) to detect C emissions (AFCS loss) between 2005 and 2015 using our forest loss data detected from the Forest/Non-Forest maps that we derived from the ESA-CCI land cover maps.
 - (2.2) Second, in lieu of our forest loss data and as suggested, we used Hansen et al. (2013) 30 m forest loss data over the same period (2005-2015) along with the same 2000 30 m AGB dataset to detect AFCS loss.
 - (2.3) Third, the results from these two assessments (one based on the ESA-CCI land cover-derived forest loss and one based on Hansen et al.' forest loss data) were compared.
 - (2.4) Fourth, we also compared the results of these two assessments with the results of Baccini et al. (2017), which for whatever reason we missed to include in our previous comparative analysis. We believe Baccini et al.'s results are credible and the most appropriate data for comparing our results with.
 - (2.5) The results of the comparative analysis are given below (Table S1).
- (3) We have kept our citation on Pearson et al. (2017) but not in the context of comparison.
- (4) Below are our revised results and discussion on the comparative analysis:

Lines 347-376 of the revised manuscript:

“Our detected C emission rate of 100 Tg C yr⁻¹ in Southeast Asia due to AFCS loss over the 2005-2015 period with the use of the ESA-CCI land cover-derived F/NF maps is close to the 118 Tg C yr⁻¹ emission rate found by Baccini et al.¹³ over the 2004-2014 period (Supplementary Table S1). And while Baccini et al.'s emission rate is slightly higher, this is expected because their estimation of C emission was performed on annual basis, thus including also the increment in biomass due to the growth of trees that occurred before deforestation. In addition, some difference is also related to the discrepancy between the 2000 AGB map that we used and the time-series AGB maps (2004-2014) that Baccini et al. used. Nonetheless, this comparative analysis provides confidence to our input data and results.

On the other hand, based on the same input AGB dataset (see Methods), the forest loss data by Hansen et al.¹⁸ over the 2005-2015 period resulted in a C emission rate of 235 Tg C yr⁻¹, more than two times higher than our finding that is based on the ESA-CCI land cover-derived F/NF maps and that of Baccini et al.¹³ (Supplementary Table S1). The large difference in quantity between Hansen et al.'s forest loss and our detected forest loss over the 2005-2015 period could explain the said large difference in C emission rates. In turn, the difference in forest loss is likely due to the interaction of the different forest definitions and spatial and temporal resolutions of the two datasets, making it difficult to disentangle the various factors.

Certainly, the relatively coarse (300 m) ESA-CCI maps may have missed fine-scale deforestation events captured instead by the high-resolution map (30 m) of Hansen et al. However, the fine-scale forest losses are mostly typical of continental Southeast Asia where about 20% of the total forest loss is detected by Hansen et al.'s dataset⁶¹, while the large-scale deforestation events occurring in insular Southeast Asia are considered detectable by the ESA-CCI maps (Supplementary Table S1). That said, where possible, the use of fine resolution data is advisable, especially in areas characterized by small-scale forest dynamics.

In a global study on greenhouse gas emissions from tropical forest degradation and deforestation between 2005 and 2010, it was found that 41% (2.6 Gt CO₂ yr⁻¹) of the emission due to deforestation came from Southeast Asia¹⁵. At the ratio of 1 Mg C is to 3.67 Mg CO₂³⁴, this translates to an emission rate of 708 Tg C yr⁻¹. But since this estimate includes not only aboveground live biomass, but also other pools including belowground live biomass, dead organic matter and soil organic matter, which have not been considered in our study, comparison could not be done. This is especially so given that most of the emissions from the peatland forests of Southeast Asia are from the soil carbon pool^{62,63}."

Baccini et al. (2017). *Science*. **358**, 230–234.

Hansen et al. (2013). *Science*. **342**, 850–853.

Pearson et al. (2017). *Carbon Balance Manag.* **12**, 3.

Country	Forest Loss				AFCS Loss					
	Based on ESA-CCI Land Cover Data (2005-2015)		Based on Hansen et al.'s Forest Loss Data (2005-2015) ¹⁸		Based on 2000 AGB and ESA-CCI Gross Forest Loss (2005-2015)		Based on 2000 AGB and Hansen et al.'s Gross Forest Loss (2005-2015) ¹⁸		^a Based on Baccini et al.'s Data (2004-2014) ¹³	
	M ha	%	M ha	%	Tg C	%	Tg C	%	Tg C	%
Brunei	0.03	0.03	0.09	0.05	0.27	0.03	1.48	0.06	0.99	0.08
Cambodia	4.00	5.03	12.50	6.36	53.24	5.34	151.36	6.44	116.11	9.88
Indonesia	49.28	61.98	120.18	61.11	619.16	62.06	1440.90	61.35	556.21	47.34
Laos	3.00	3.77	7.10	3.61	38.25	3.83	80.77	3.44	85.93	7.31
Malaysia	13.18	16.57	37.03	18.83	173.51	17.39	470.94	20.05	184.11	15.67
Myanmar	4.18	5.26	7.46	3.80	46.01	4.61	80.55	3.43	113.28	9.64
Philippines	2.41	3.04	0.90	0.46	24.55	2.46	11.22	0.48	24.49	2.08
Singapore	0.01	0.01	0.00	0.00	0.04	0.00	0.02	0.00	No Data	No Data
Thailand	0.59	0.74	2.88	1.46	6.00	0.60	28.45	1.21	32.34	2.75
Timor-Leste	0.02	0.03	0.02	0.01	0.21	0.02	0.19	0.01	0.63	0.05
Vietnam	2.82	3.54	8.50	4.32	36.50	3.66	82.80	3.53	60.87	5.18
SE Asia	79.51		196.67		997.73		2348.69		1174.96	100.00
Annual	7.95		19.67		99.77		234.87		117.50	

^a These values are average values over the 2004-2014 period derived from Table S1 of the cited source.

- Forest/forest change patterns on maps (Fig. 1 and supplementary Fig. 1) differ from own experiences and observations of satellite imagery substantially. Example 1 Cambodia (Fig. 1a and supplementary Fig. 1): too little deforestation in the Boeng Peae Wildlife Sanctuary in the 2015 situation and therefore basically no modeled forest loss in that area for 2050 (which is by now in 2018 already to a large extent deforested). Example 2 Borneo (Fig. 1c and supplementary Fig. 1): main modeled deforestation occurs in areas in the north of Sarawak along the coast where there seems to be not much forest left in 2015 (maybe map scale too low). However, along the southeastern border of the forest there is even under the worst scenario SSP 3 no further forest loss expected, even though there was quite a substantial amount of deforestation detected in 2015. This as well could be an indication that the used input data (ESA-CCI maps) might be not appropriate to reliably depict smaller-scale deforestation patterns.

RESPONSE: Thank you very much for the comments. We have improved the figures (Fig. 1; Supplementary Fig. S6). The projected deforestations in Cambodia and Borneo are now clearly visible. There are also some projected deforestations on the north-eastern side of Boeng Peace Wildlife Sanctuary.

For the quantity of future forest cover loss under SSP 3 (now also under SSP 5, with the inclusion of the other 3 SSPs), it is dependent on the quantity that we obtained from our downscaling procedure (Supplementary Fig. S2a) and the raw quantity under each SSP taken from the SSP database (Supplementary Fig. S1). For the location of future deforestation or forest loss, it was based on the derived transition potential maps (Supplementary Figs. S3 and S4) with an average SM value of 0.56.

- When looking in detail at Fig. 1, the modeled map does not indicate any change in several countries where there is – according to known trends and statistics – substantial change ongoing (e.g. deforestation statistics in Myanmar and Laos, see FAO FRA).

RESPONSE: Thank you for the comment. The figures have been improved and the projected deforestations are now clearly visible (Fig. 1; Supplementary Fig. S6). Figures 2 and 4 present the quantities of deforestation and AFCS loss under SSP3 across countries, including Myanmar and Laos, and Table S4 presents all quantities of forest cover and AFCS changes across all countries and SSPs.

Besides above major concerns I have some further points that need clarification:

- The analysis of the ESA-CCI maps as input for their models let the reader deduce that the same forest definition (as for the ESA-CCI maps) must have been applied in this study. However, it would be good if the authors could clearly describe their applied forest definition as it is not explicitly mentioned in their manuscript.

RESPONSE: Thank you very much for pointing this out. We completely agree and thank you for the suggestion for us to be explicit. But since our derived Forest/Non-Forest maps are output of a reclassification procedure, we have defined forest as follows:

Lines 443-449 of the revised manuscript:

“The land cover maps have a thematic resolution of 37 land cover categories, which we reclassified into two categories, namely forest and non-forest, following the reclassification procedure used in Estoque et al.⁷³. Thus, forest is herein defined as lands classified under any of the following categories as per ESA-CCI land cover dataset (v2.07): (class codes 50, 60-62) tree cover, broadleaved, evergreen and deciduous, closed to open; (class codes 70-72, 80-82) tree cover, needleleaved, evergreen and deciduous, closed to open; (class code 90) tree cover, mixed leaf type (broadleaved and needleleaved); and (class codes 160, 170) tree cover, flooded, fresh or brackish water, saline water. The resulting maps are herein called forest/non-forest (F/NF) maps.”

- Regarding the modeled gains in forest cover, it would be very interesting to know for some example countries what could cause that gain under SSP 1 (e.g. forest plantations, abandoned areas, etc.)? I personally would like to know that in the case of Malaysia as I cannot imagine where and how this could take place (is it realistic that plantation areas will be abandoned – especially with the expected elevated resource-demands of a growing human population)?

RESPONSE: Thank you very much for the comment and questions. We acknowledge that, as this is about future, it is uncertain how the SSPs' projected forest gains in Malaysia could be achieved. Under SSP 1, its storyline is that land use is strongly regulated, and tropical deforestation rates are greatly reduced. It also assumes inclusive development and respect for perceived environmental boundaries, as well as high investment on human capital, education and awareness.

Nevertheless, this issue has been included in our discussion.

Lines 310-326 of the revised manuscript:

“However, it should be stressed that for the projected forest cover gains to be achieved, especially under SSP 1, effort should focus on both the protection and conservation of the remaining forests and the expansion of forest cover through reforestation and/or afforestation. Furthermore, we recognize the importance of tree plantations for economic purposes across the region, more importantly in Malaysia and Indonesia, but tree plantations for ecological purposes must also be considered. To cite one example, the Philippines' 2011 forest rehabilitation initiative, called the National Greening Program (NGP), which initially aimed to plant 1.5 billion trees on 1.5 million ha by 2016, is a government priority program to reduce poverty, promote food security, environmental stability and biodiversity conservation, and enhance climate change mitigation and adaptation^{56,57}. The program has been expanded (ENGP; 2016-2028) with the aim to reforest all remaining unproductive, denuded, and degraded forestlands nationwide of about 7.1 million ha which also contribute to environment-related risks such as soil erosion, landslides, and flooding⁵⁸. Perhaps, at this stage and in relation to SSP 1, what is important is for individual countries to formulate, if not yet available, such kind of initiative that also considers the future environmental goal of tree planting or re-greening (i.e. reforestation and/or afforestation for biodiversity and regulating and supporting services), not only focusing on its economic prospect (i.e. tree planting for a near-future harvest of provisioning services). In this regard, a strong policy support is needed.”

- In this context I would like to ask the authors to resize both figures 1 (of the main manuscript as well as in of the supplementary data) and the sub-figures to be split into separate figures at reasonable scale to be able to allow the reader to evaluate better the derived maps.

RESPONSE: Thank you very much for the comment and suggestion. The figures have been improved (Fig. 1; Supplementary Fig. S6).

- In the manuscript it would be better if the authors could clearly explain (already in the results section) why they also analyzed IF and PA areas? Right now, these results are described next to the total forest figures without any explanation, leading most probably to some confusion of the reader.

RESPONSE: Thank you very much for the comment and suggestion. We have revised the text to consider this comment, as follow:

1) Within the two introductory sub-sections, we have discussed the importance of IFs and PAs.

2) We have also clarified our research goal:

“Here, we spatially allocated the projected future forest cover changes under the five baseline SSPs by employing a state-of-the-art land change modeling approach and remotely sensed data (2015-2050). We examined the potential implications of these spatially allocated forest cover changes by quantifying their consequent AFCS changes at the country and province levels, across forest cover classes, and within the IFs and PAs in Southeast Asia, given their important roles as prime reserves for tropical biodiversity and AFCS (see Methods).” (Lines 83-88 of the revised manuscript)

3) In the Methods section, the last sub-sections is about “Estimating future AFCS losses across forest cover classes and within IFs and PAs (2015-2050)”.

4) Finally, the Results section has been re-organized as follows:

- Past-to-present forest cover and AFCS losses in Southeast Asia

- Projected forest cover and AFCS changes in Southeast Asia across the five SSPs
- Projected forest and AFCS losses within Southeast Asia's forest classes, IFs and PAs
- Model validation

- At least in the supplementary data the authors could consider adding a map of IF and PA at reasonable scale. This could be very informative for the reader.

RESPONSE: Thank you very much for the suggestion. It has been followed. Please see Supplementary Fig. S8 of the revised manuscript.

- In this context, the two results sections (lines 101-142) describe in detail the various percentage values of forest cover gain or loss per country. This should better be condensed in one or two tables.

RESPONSE: Thank you very much for the suggestion. The texts citations of numbers and percentages have been reduced. The full statistical information on quantities is given in the Supplementary Materials section.

- Within the discussion section (Comparison, Uncertainties and Insights), results of ESA-CCI figures and FRA FAO records are compared to each other. Such kind of direct comparison is, however, not directly feasible because the former show changes in 'forest cover', while the latter elaborate on 'forest land' ('land use'), which is not the same.

RESPONSE: Thank you very much for the comment. We have carefully revised and improved the entire section labelled "Comparison, Uncertainties and Insights.", and we have stated that the FAO FRA statistics could not be used as a source of comparison due to various reasons, one of which being the difference between the "land cover" definition of forest employed in the datasets used for our estimates and the "land use" definition of forest employed by FAO (Please see Lines 334-345 of the revised manuscript).

Reviewer #2 (Remarks to the Author):

As my expertise is specifically in land change modeling, and not in scenarios of forest change and their implications, my comments are focused almost exclusively on the land change modeling process. The authors are to be commended on an effective use of the technology deployed, and the model skill exhibited is good.

RESPONSE: We sincerely thank Reviewer #2 (Prof. Eastman) for his openness, time, effort and expertise. We sincerely appreciate it. Please find below our point-by-point response.

On lines 424-426, they state “Using information from training samples, MLP NN performs a non-parametric regression analysis between input variables and one dependent variable with the output containing one output neuron, i.e. the predicted memberships”. This is not strictly correct. MLP actually has two output neurons for each transition modeled. For example, if the transition being modeled is forest to non-forest, a sample is taken of pixels that experienced that transition during the historical period (change), and an equal size sample of pixels that were eligible to change, but which did not (persistence). In the implementation in LCM, the user doesn’t see the persistence class, but it is there. As a result, the procedure used is not a regression, but rather, a classification. The transition potential that the user sees is the activation level of the output neuron of change. This is indeed a probability (see lines 427-428), but a special one – it is the probability of change IF the prior probability of change were 0.5. It has this prior probability because the system used a balanced sample (equal samples of change and persistence). Because of this special character, transition potentials are only used in a relative sense when allocating predicted change.

RESPONSE: Thank you very much for these important clarifications. We have revised and clarified our texts as follows:

Lines 500-509 of the revised manuscript:

“The whole part of the transition potential modeling (Supplementary Fig. S2) included the processes of training and testing using sample pixels that were randomly selected by the LCM’s MLP NN (Supplementary Table S7). For instance, for the transition ‘from forest to non-forest’, a sample of equal size is randomly selected from both the pixels that experienced forest loss and those pixels that were eligible to change but did not (persistence). Of the total sample selected, a half (50%) is used for training, while the other half (50%) is used for testing. Based on the training samples and the resulting transition potentials of the pixels, the model performs a F/NF classification, whose results are compared against the testing samples (see Model Validation section below). The pixel values of the transition potential maps (TPMs) (Supplementary Figs. S3 and S4) are considered probability of change given that the prior probability of change is 0.5 because of the equal size of samples from the changed pixels and non-changed pixels (persistence). More details can be found in TerrSet’s documentation and help system on LCM’s MLP NN.”

The sentence on lines 429-431 is effectively true, but be aware that behind the scenes, there were also transition potentials generated for the two persistence classes (which were discarded, since they were unnecessary for the allocation stage of the prediction).

RESPONSE: Thank you very much for this. It has been noted and we keep this in mind every time we use LCM.

Line 476/477 is confusing and could benefit from some clarification. Mercator is not an equal area projection. How then were areas calculated?

RESPONSE: Thank you very much for pointing this out. Our apologies for the confusion. To make sure we got our fundamental pre-processing procedures right, we have redone all the pre-processing procedures and re-run all the simulations. All data were projected to the “Asia South Albers Equal Area Conic” projection system. The area of current forest cover and the projected forest cover changes changed slightly. The Skill Measure values have also changed a little in each country, but the overall average still rounds to 0.56.

I hope this helps.

RESPONSE: Yes, certainly it did help. Again, thank you very much.

Ron Eastman

Reviewer #3 (Remarks to the Author):

This paper describes an effort to spatially explicitly simulate forest cover transitions in South East Asia under two scenarios. The authors used readily available remote sensing data to detect forest cover loss and gain, seven static spatial determinants to calculate transition probability and a land change model to simulate the changes up to 2050. In addition, they link their forest cover gain and loss with aboveground forest carbon stock. Overall, the paper is written in fluent English, the context of the approach is well defined and the topic is important and merits to receive attention.

RESPONSE: We sincerely thank Reviewer #3 for the compliments, expert comments/suggestions, and the time and effort in reviewing our manuscript. We sincerely appreciate it. Please find below our point-by-point response.

My main concerns are with the rationale of the methodological strategy and are appended below:

I agree with the authors that spatial allocation and mapping of projected large-scale quantities of forest cover change are important for many reasons. What puzzled me (and potentially will be puzzling to the readers) is that, on the one hand, in the Asian region, increasing trends are shown in SSPs 1, 2, 4 and decreasing trends in SSPs 3 and 5 (lines 347-349). On the other hand, they choose to simulate and consider, as the most plausible, the two scenarios with the opposing ends of the spectrum (line 75). If the authors performed the analysis for all five scenarios and they chose to present only the two contrasting, I would advise presenting the full range of spectrum as I believe the paper will benefit greatly (and is much more aligned with the title of this paper). Not only it would capture the full range of plausibility but also the potential implications of each pathway would be better pronounced.

RESPONSE: Thank you very much for the comment/suggestion. We completely agree that it would be more informative to include the full scenario spectrum. Thus, the suggestion has been followed. We have included all five SSPs in the revised manuscript. The full results for all SSPs in terms of forest cover change are given in Fig. 1 and Supplementary Table S4, and the full results for all SSPs in terms of AFCS loss are presented in Fig. 3 and Supplementary Table S4. SSPs 1 and 3 are highlighted in Figs. 2 and 4.

Specific comments

Line 67: is there any measurement of plausibility for Asia? How these two extremes are equally plausible?

RESPONSE: Thank you very much for pointing this out. Our apologies for the confusion. In our usage of the phrase “equally plausible”, we meant to say that both scenarios can happen, without the intention of giving them equal magnitude or possibility of happening. But since this is potentially confusing and with the inclusion of the other 3 SSPs, as suggested, we have dropped the phrase “equally plausible” in the revised manuscript.

Line 94: What does the overall average of 0.56 mean?

RESPONSE: Thank you very much for pointing this out. This has been clarified in the revised version as follows:

Lines 149-161 of the revised manuscript.

“Our modeled forest transition potential maps (TPMs) (Supplementary Figs. 3 and 4) played a key role in the spatial allocation of the projected future forest cover changes. Here, the Skill Measure (SM) statistic was used to assess the predicted power of these TPMs (see Methods). This statistic is a robust validation parameter because it compensates for the dependence of expected accuracy on the number of transitions and persistence classes. In this study, the SM values derived for the 11 countries in Southeast Asia had an overall average of 0.56 (Supplementary Table S8). Considering that this overall average SM value is well above 0 (a case where the model is no better than chance), but more especially above -1 (a case where the model is worse than random chance), our modeled TPMs have predictive power to simulate correctly the past-to-present spatial patterns of forest cover changes in the region, which are likely to continue into the future. Needless to say, the transition potential modeling was performed at a large scale, i.e. country level. Generally, our derived SM values are either close to or within the range reported in other related studies³¹⁻³³.”

Figure 1 needs some improvements e.g countries name to help the reader navigate and understand the geographical context.

RESPONSE: Thank you very much for the suggestion. Fig. 1 has been improved. Country boundary layer has been added.

Figure 2 needs to be provided in better resolution.

RESPONSE: Thank you very much for the suggestion. Fig. 2 has been improved. Also, all figures have been submitted as separate individual image files in high resolution format.

Lines 416-431/Transition potential modeling: The authors used 2005-2015 changes to train a Neural Network model that in turn created the transition potential surfaces. I believe their methodology suffers from two limitations that have a significant effect on the results.

First, normally in land change modeling and especially in the transition potential modeling part, it is feasible to detect changes for a period of time and incorporate in the model several aspects of these changes using environmental and socio-economic determinants that explain these changes. The most meaningful way to do so, is to detect changes eg for this case 1992-2005, use as many determinants useful to explain these changes, create the transition potential surface, simulate up to 2015 and compare the simulation product vs the reality (reclassified map of 2015). This calibration process must be repeated until a level of accuracy is reached and is crucial the performance capability of a model.

RESPONSE: Thank you very much for the comments.

- 1) We completely understand the point raised.
- 2) We recognize that in the land change science field, a field from which some of us authors are also coming from (including the lead author), the processes of calibration and validation are important components of a land change modeling exercise.
- 3) Please allow us to expound the context. For a pure predictive modeling, the change between t1 (1992) and t2 (2005) maps along with a set of spatial driver variables are used for calibration (for both quantity and spatial location of change), and the change between t2 (2005) and t3 (2015) maps are used for validation. Thus, the validation outputs are composed of error due to quantity and error due to allocation, which are often expressed as hits, misses and false alarms. Here, predictive modeling refers to a type of modeling that is calibrated based on information from past changes, aimed at predicting future changes.
- 4) We could have used t1 (1992) and t2 (2005) maps, as pointed out, to calibrate a transition potential map and set the quantity of change to be equal to the actual change between t2 (2005) and t3 (2015) so that the validation results (simulation errors) would only be pertaining to error due to allocation. This is because, in this study, we are considering future scenarios in which the quantity of change is already determined. However, we realized that the information coming from the land change (forest loss and gain) that occurred between t1 (1992) and t2 (2005) are themselves important driver variables for future land change (forest loss and gain): (i) distance to deforested or reforested area (1992-2005); and (ii) forest cover loss or gain share per province (1992-2005). Had we used t1 (1992) and t2 (2005) maps as inputs, we would have excluded these variables because of data availability problem for time periods prior to 1992. Excluding them could have produced transition potential maps that are much inferior than what we currently have.
- 5) In other words, we have used land change-related information from the t1 (1992) and t2 (2005) maps and transformed them as spatial driver variables. These variables played a key role in the calibration of our current transition potential maps with a Skill Measure that is well above 0 (random chance).

- 6) Our approach, which makes use of the t2 (2005) and t3 (2015) maps because the information on land change that occurred between time t1 (1992) and t2 (2005) have already been used as spatial driver variables, takes advantage of the important feature of LCM, i.e. being able to carry out training (calibration) and testing (validation) by dividing the sample pixels into two sets: 50% for training and 50% for testing (Please see the following sub-sections under the Methods section of the revised manuscript: Transition Potential Modeling; and Model Validation).
- 7) Thus, despite all these limitations, we are confident that our approach is methodologically sound and acceptable. This being said, we would welcome and sincerely appreciate any further comments and suggestions. Thank you very much.

Second, and equally important, by using actual 1992-2015 forest cover loss as a determinant you introduce to your model a large amount of bias.

RESPONSE: Thank you very much for the comment.

To our knowledge, and please correct us if we are wrong, there is still no available methodology to downscale the projected forest cover changes under the SSPs down to country level. Therefore, we used the forest cover change information from the past to present (1992-2015) to proportionally allocate these projected forest cover changes (2015-2050) to each country. In effect, our approach to calibrating quantity of change is also a hybrid between a predictive modeling (using past information) and a scenario modeling (using scenario-based projected quantities of forest cover changes). As the comment is short, we could not completely grasp and comprehend how this said approach introduces large amount of bias. Any further comments and suggestions would be sincerely appreciated.

Lines 422-423: It is not clear what procedures were followed per scenario. Given that two contradicting scenarios were chosen, this paper only uses static variables to spatially determine a complexity. How the driver variables differ per scenario? I suspect that population, economics and other proxies are important determinants as well and should variate accordingly per scenario (as the authors also mention in lines 218-230).

RESPONSE: Thank you very much for the comments.

As we have clarified above, our transition potential modeling is a predictive modeling, in which information on the past spatial pattern of forest cover changes (loss and gain) are used to predict potential spatial locations of future forest cover changes. We used distance to urban area and roads as proxies for socioeconomic variables, including population.

Population and GDP are the prime socioeconomic variables projected under the SSP framework, and they too, need to be downscaled. However, downscaling of these variables is also one of the important current issues today. And it is not the focus of our study. Nonetheless, this issue has been pointed out in our discussion (see also below).

Lines 385-396 of the revised manuscript

“There is also a need to downscale the SSP’s projected socioeconomic variables, e.g. population and gross domestic product⁶⁴⁻⁶⁶, to enable researchers to examine how these projected variables would relate to the spatial pattern of future forest cover changes. In such a case, a forest cover change scenario modeling can be conceptualized and be based purely on projected quantities and spatial locations of future forest cover changes.”

The idea is that, once these variables have been downscaled, they can be examined whether they can indicate or help determine the spatial locations of future forest cover changes. In such as case, there

would not be a need to do transition potential modeling because the potential locations of future forest cover changes would be purely based on assumptions and scenario storylines. As a result, the entire land change modeling process can become entirely a type of scenario modeling, i.e. without considering any information from past land changes. An example of this is the study of Kubiszewski et al. (2017; *Ecosyst. Serv.* **26**, 289–301), in which both their quantity and spatial locations of future land changes are based on scenarios and assumptions. We reviewed this study (Kubiszewski et al. 2017) in our paper in the Methods section.

Line 424: How many samples, per country or province. Also, it is important to mention the sampling strategy (eg random, stratified, weighted in changed areas etc).

RESPONSE: Thank you very much for pointing this out. The requested information are now added to the text. The selection of samples was done by the LCM employing a random selection method. The quantity of samples per country is listed in Table S6 of the revised manuscript.

Lines 500-504 of the revised manuscript:

“The whole part of the transition potential modeling (Supplementary Fig. S2) included the processes of training and testing using sample pixels that were randomly selected by the LCM’s MLP NN (Supplementary Table S7). For instance, for the transition ‘from forest to non-forest’, a sample of equal size is randomly selected from both the pixels that experienced forest loss and those pixels that were eligible to change but did not (persistence). Of the total sample selected, a half (50%) is used for training, while the other half (50%) is used for testing.”

Lines 443-452/Model Validation: This part is not clear as it currently stands. Which are the inputs for calculating SM? I suspect that this step is affected.

RESPONSE: Thank you for the comment. The Model Validation part has been clarified:

Under Methods section: Lines 522-531 of the revised manuscript.

Model Validation. The MLP NN algorithm in LCM outputs a statistic called Skill Measure (SM) which indicates the predicted power of the TPMs (Supplementary Figs. S3 and S4). SM is a robust validation measure because it compensates for the dependence of expected accuracy on the number of transitions and persistence classes. It is calculated as (Eq. (1)) (TerrSet’s Help System):

$$SM = \frac{A - E(A)}{1 - E(A)} \quad (1)$$

where A is the measured accuracy based on a confusion matrix. E(A) is the expected accuracy, expressed as: $E(A) = 1/(T + P)$, where T is the number of transitions in the model and P is the number of persistence classes or the number of ‘from’ in the sub-model. SM has a value ranging from -1 (worse than chance) to +1 (perfect prediction), where zero indicates ‘no better than chance’. Our model validation results are given in Supplementary Table S8.”

Under Results section: Lines 149-161 of the revised manuscript

“Our modeled forest transition potential maps (TPMs) (Supplementary Figs. 3 and 4) played a key role in the spatial allocation of the projected future forest cover changes. Here, the Skill Measure (SM) statistic was used to assess the predicted power of these TPMs (see Methods). This statistic is a robust validation parameter because it compensates for the dependence of expected accuracy on the number of transitions and persistence classes. In this study, the SM values derived for the 11 countries in Southeast Asia had an overall average of 0.56 (Supplementary Table S8). Considering that this overall average SM value is well above 0 (a case where the model is no better than chance), but more especially above -1 (a case where the model is worse than random chance), our modeled TPMs have predictive power to simulate correctly the past-to-present spatial patterns of forest cover changes in the region, which are likely to continue into the future. Needless to say, the transition potential modeling was performed at a large scale, i.e. country level. Generally, our derived SM values are either close to or within the range reported in other related studies³¹⁻³³. ”

REVIEWERS' COMMENTS:

Reviewer #1 (Remarks to the Author):

I would like to congratulate the authors for their interesting work! I am satisfied with the modifications the authors made relating to my former comments and I have no objections regarding a publication.

Andreas Langner

Reviewer #2 (Remarks to the Author):

I have reviewed the changes made concerning the land change modeling component, and I am satisfied with the methods used and the way they are described in the article.

Reviewer #3 (Only left remarks to Editor)

REVIEWERS' COMMENTS:

Reviewer #1 (Remarks to the Author):

I would like to congratulate the authors for their interesting work! I am satisfied with the modifications the authors made relating to my former comments and I have no objections regarding a publication.

Andreas Langner

RESPONSE: We sincerely thank Reviewer 1, Dr. Andreas Langner, for his expertise, and time and effort in reviewing our manuscript. We sincerely appreciate it.

Reviewer #2 (Remarks to the Author):

I have reviewed the changes made concerning the land change modeling component, and I am satisfied with the methods used and the way they are described in the article.

RESPONSE: We sincerely thank Reviewer 2, Prof. Ronald Eastman, for his expertise, and time and effort in reviewing our manuscript. We sincerely appreciate it.

Reviewer #3 (Only left remarks to Editor)

RESPONSE: We sincerely thank Reviewer 3 for his expertise, and time and effort in reviewing our manuscript. We sincerely appreciate it.